

# The Risk of Synoptic-Scale Arctic Cyclones to Shipping

Alexander F. Vessey[1,2], Kevin I. Hodges[2,3], Len C. Shaffrey[2,3], and Jonathan J. Day[4]

[1]AXA XL, 20 Gracechurch Street, London, UK, EC3V 0BG
[2]Department of Meteorology, University of Reading, Earley Gate, Reading RG6 6BB, UK
[3]National Centre for Atmospheric Science, University of Reading, Earley Gate, Reading RG6 6BB, UK
[4]ECMWF, Shinfield Park, Reading RG2 9AX, UK

**Correspondence:** Alexander F. Vessey (alecvessey@hotmail.co.uk)

**Abstract.**

The risk posed by Arctic cyclones to ships has seldom been quantified due to the lack of publicly available historical Arctic ship track data. This study investigates Automated Identification System (AIS) transponder derived Arctic ship tracks from September 2009 to December 2016. These are analysed with historical synoptic-scale cyclone tracks derived from ERA-5 and reports of past Arctic shipping incidents, to determine the number of ships intersected by the passage of intense Arctic cyclones, and how many resulted in shipping incidents.

The number of ships operating in the Arctic has increased year-on-year from 2010 to 2016. The highest density of ships occurs year-round in the Barents Sea. Trans-Arctic shipping transits via the Northern Sea Route and the North-West Passage are limited to summer and autumn months, when sea ice extent has sufficiently retreated from the coastlines. But, ship traffic along these trans-Arctic routes is far less than the thousands of ships travelling in the Barents Sea year-round. Between 2010 and 2016, 248 Arctic shipping incidents were reported, but only 2% of these occurred following the passage of an intense Arctic cyclone. So, shipping incidents do occur in the Arctic, but the vast majority appear unrelated to the passage of intense Arctic cyclones, despite ship tracks being frequently intersected by such hazards. Less than 0.0001% of these intersections resulted in a reported shipping incident. This suggests that Arctic cyclones have not been hazardous to ships, and that ships are resilient to the rough sea conditions caused by intense Arctic cyclones.

## 1 Introduction

As a consequence of global warming, The Arctic Ocean is becoming increasingly accessible for ships as Arctic sea ice continues to decline (Stroeve et al., 2007, 2012, 2014). In September 1980, Arctic sea ice extent was 7.7 million km$^2$, but was only 3.6 million km$^2$ in September 2012 (National Snow & Ice Data Centre, 2023). Arctic sea ice is projected to decline further into the future as global surface temperatures are projected to increase further (Stroeve et al., 2012; Wei et al., 2020). This reduction in Arctic sea ice extent does however provide opportunities for industries such as shipping, oil exploration, and tourism in the Arctic, which include shorter journeys between ports in North America, Europe and Asia (Smith and Stephenson, 2013; Melia et al., 2016, Table S1 in Supplementary Material), access to previously inaccessible natural resources (Harsem et al., 2015), and new destinations for tourism (Maher, 2017). As a result, it is expected that the number of



ships travelling in the Arctic has and will continue to increase the exposure to extreme weather (Browse et al., 2013; Lasserre, 2014; Melia et al., 2016; Lasserre, 2019).

Despite this dramatic reduction in Arctic sea ice extent over the past few decades, the trans-Arctic shipping routes of the Northern Sea Route (NSR) and North-West Passage (NWP) only recently became open simultaneously in 2008 (European Space Agency, 2008). The NSR follows the coastline of Eurasia and connects Europe and Asia when Arctic sea ice has

sufficiently retreated northward from the Eurasian coastline. The NWP runs through the Canadian Archipelago when free of sea ice and connects North America with Asia. These shipping routes can provide shorter journeys between major ports in these continents than the more traditional and tropical Suez Canal and Panama Canal routes (Table S1 in Supplementary Material). For example, the distance along the NSR is approximately 40% shorter than the Suez Canal route between Europe and Asia (Schøyen and Bråthen, 2011, Table S1 in Supplementary Material). Since 2008, these trans-Arctic shipping routes have become

ice-free for longer periods in summer and autumn months (Melia et al., 2016), as Arctic sea ice extent has continued to decline (National Snow & Ice Data Centre, 2023).

But, the Arctic is a challenging environment for such human activity. Cold temperatures can make working conditions difficult and can cause equipment failures (Larsen et al., 2016), and sea ice can also confine ships to travel over the shallow and perilous coastlines around the boundaries of the Arctic Ocean (Arctic Monitoring & Assessment Programme: Working

Group of the Arctic Council, 2020). Conditions can be made even more dangerous by the passage of a cyclone, which can cause rough sea conditions due to high winds and high ocean waves (Thomson and Rogers, 2014; Liu et al., 2016; Waseda et al., 2018, 2021). Such conditions could potentially capsize a shipping vessel and its cargo and endanger its crew, or cause delays in transit. Arctic cyclones can also enhance the break-up of sea ice (Simmonds and Keay, 2009; Asplin et al., 2012; Parkinson and Comiso, 2013; Peng et al., 2021), which can drift into shipping lanes and become an additional hazard for ships

to navigate.

Some recent Arctic shipping disasters highlight how perilous the Arctic can be for shipping. On 23 March 2019, the MV Viking Sky cruise ship, with 1,373 people on board, lost power whilst trying to contend with extreme wind and ocean wave conditions caused by the passage of an Arctic cyclone (Ibrion et al., 2021). Waves were reported to be in excess of 15 metres in height. Passengers and crew were rescued using helicopters and rescue boats, as the cruise ship drifted toward shallow

coastlines of Norway. After a full evacuation, the cruise ship was salvaged, but some damage had to be repaired and imminent trips were cancelled. Other Arctic cruise ship incidents unrelated to the passage of a cyclone have been documented, such as the MV Akademik Ioffe ship running aground On the $24^{th}$ August 2018 and spilling 80 litres of fuel (Transportation Safety Board of Canada, 2018; Johannsdottir et al., 2021), and the MV Clipper Adventurer being damaged whilst running aground in on $27^{th}$ August 2010 (Johannsdottir et al., 2021). These case studies highlight that ships are at risk when travelling through the

Arctic. However, no comprehensive study has yet determined how many ships may be at risk to Arctic cyclones.

The spatial distribution and intensity of Arctic cyclones varies seasonally, with the highest density of winter (DJF) Arctic cyclones typically occurring over the Norwegian, Greenland and Barents seas, and the highest density of summer (JJA) cyclones typically occurring over the coastline of Eurasia and the Arctic Ocean (Reed and Kunkel, 1960; Serreze et al., 2001; Simmonds et al., 2008; Crawford and Serreze, 2016; Vessey et al., 2020). Vessey et al. (2022) showed that this spatial





distribution is also shown in the maximum intensity locations of the most intense winter and summer Arctic cyclones. Arctic cyclones in winter are also generally more intense than summer Arctic cyclones (Zhang et al., 2004; Sorteberg and Walsh, 2008; Simmonds et al., 2008; Vessey et al., 2020, 2022). Although synoptic-scale Arctic cyclones have been the focus of many studies in the past, the exposure of ships to intense Arctic cyclones has seldom been reported on.

In December 2004, it became mandatory for all large ships with a gross tonnage greater than 300 tonnes and all passenger ships regardless of their size to have Automated Identification System (AIS) transponders, which transmit the ships' location to satellites in real time (International Maritime Organization, 2020). This regulation was established by the International Maritime Organization to increase the safety of ships in often busy shipping lanes. Due to their safety benefits, AIS transponders have been increasingly fitted to smaller vessels, and in May 2012, it became mandatory for all fishing vessels with a size greater than 24 metres to have AIS transponders (U.K. Gov., 2014). This has allowed for the monitoring of ships and recording of past ship tracks. However, archived historical ship track data is often privately owned and difficult or costly to obtain publicly.

Consequently, there are few publicly available studies that describe past Arctic shipping activity, likely due to the lack of Arctic shipping datasets (e.g., Corbett et al., 2010; Eguiluz et al., 2016; Hreinsson, 2020; Berkman et al., 2020b, 2022). These studies show that there is typically a high density of ships in the Barents Sea year-round, and that trans-Arctic shipping along the NSR and the NWP is currently limited to months where sea ice extent is near its minimum, typically from August to October. But, these studies do not combined information of past Arctic cyclone tracks with historic ship tracks to assess the risk that Arctic cyclones pose to shipping. It is therefore unclear whether the number of shipping incidents in the Arctic are increasing as the Arctic becomes more accessible due to declines sea ice extent, how many ships are presently at risk to hazardous weather conditions caused by Arctic cyclones, and how many Arctic shipping disasters have occurred following the passage of cyclones.

The lack of publicly available historic ship track data has been somewhat been addressed by Berkman et al. (2020a), who published an open-source Arctic ship track dataset. This contains the transmitted AIS ship location data of ships that travelled north of the Arctic Circle (north of 66.3°N), but is only available for a limited period from September 2009 to December 2016. Berkman et al. (2020b, 2022) used this dataset and showed that the number of ships in the Arctic has increased between 2010 and 2016. Combining past ship tracks with past cyclone tracks could provide new insights into quantifying the risk of cyclones to Arctic shipping.

This study aims to describe Arctic shipping activity and incidents between September 2009 and December 2016, and quantify the number of Arctic ships that have been intersected by intense cyclones, using publicly available shipping data. This will be achieved by answering the following research questions:

– Given recent reductions in Arctic sea ice extent, is there evidence of any trend in the number of Arctic shipping incidents?

– Does the spatial distribution of Arctic shipping vary with the seasonal changes in Arctic sea ice extent?

– How many Arctic ships have been intersected by past intense cyclones, and how many of these led to a reported shipping incident?



The methods used in this study are described in Section 2, including a description of the data and storm tracking method used. In Section 3, the results from his study are described, detailing the trends and seasonal spatial distribution of Arctic shipping, the frequency and spatial distribution of intense Arctic cyclones, and how intense Arctic cyclones impact ships. Finally, a summary of the main conclusions is given in Section 4.

## 2 Methodology

### 2.1 Historic Atmosphere and Ocean Data

This study uses atmospheric data from the most recent reanalysis dataset from the European Centre for Medium Range Weather Forecasts (ECMWF), ERA-5 (Hersbach et al., 2018, 2020). Reanalysis datasets have been developed over recent decades to provide an accurate 4-dimensional representation of past atmospheric conditions, created by assimilating historical observations from a range of sources into state-of-the-art Numerical Weather Prediction (NWP) models. Although there are multiple reanalysis datasets available from various institutions, ERA-5 was chosen here as it is the most recent and highest spatial and temporal resolution of all reanalysis datasets available (Vessey et al., 2020).

ERA-5 contains atmospheric data from 1940-present at a 1-hourly temporal resolution and at an approximately 31km (TL639) spatial resolution, with 137 vertical levels up to 0.01 hPa. Historical observations are assimilated into the ECMWF Integrated Forecast System (IFS) version CY41R2, using a 4-dimensional variation data assimilation scheme (Hersbach et al., 2020). Prior to 1979, satellite observations were not available, so the reanalysis datasets may be less constrained. Thus, in this study, data from ERA-5 is used from 1979-2021 to assess the climatology of intense Arctic cyclones.

The ERA-5 atmospheric variables used in this study are the 850 hPa relative vorticity and 10-metre u- and v- component winds. The IFS model is also coupled to the ECMWF WAM (WAve Model) Model and gives information of past ocean states, but at a lower spatial resolution of 0.5°. The ECMWF WAM Model is able to determine past wave heights over the open ocean, but cannot capture waves within sea ice. To assess how Arctic cyclones influence the ocean state and cause hazardous rough sea conditions, the ERA-5 significant wave height including tide and surge field is also used. These ERA-5 variables are used at 1-hourly intervals each day.

In this study, historic Arctic shipping activity is also related to past Arctic sea ice extent. For this purpose, the Met Office Hadley Centre Sea Ice and Sea Surface Temperature version 2.1 dataset (HadISST2.0) (Titchner and Rayner, 2014) is used to indicate past Arctic sea ice extent. This dataset was created by combining various Arctic sea ice records to produce a best-estimate of past sea ice extent globally at a 1° horizontal resolution from 1850 to present (Titchner and Rayner, 2014).

### 2.2 Storm Tracking

Arctic cyclones are identified in hourly ERA-5 data using the storm tracking algorithm developed by (Hodges, 1994, 1995, 1999, 2021), which has been used in various studies to identify past Arctic cyclones in reanalyses (e.g., Day and Hodges, 2018; Day et al., 2018; Gray et al., 2021; Vessey et al., 2022). Vessey et al. (2020) showed that this storm tracking



algorithm captures more Arctic cyclones when based on 850 hPa relative vorticity than mean sea level pressure (MSLP), so in
this study 850 hPa relative vorticity is used as the storm tracking variable.

This field is first spectrally truncated to a spectral resolution of T42 and is filtered to remove the planetary scales for total
wavenumbers less than or equal to five. This ensures that synoptic-scale systems that are independent of large-scale forcings
are focused upon. Cyclone features are then identified at each timestep as maxima in the T42 850 hPa relative vorticity field.
Feature points between consecutive hourly timesteps within a minimum displacement factor of 2° in all regions north of 30°N,
are then linked into create cyclone tracks. This is achieved by optimising a cost function for track smoothness, which is subject
to adaptive constraints on displacement and smoothness (Hodges, 1999).

Once all cyclone tracks have been identified between 1979-2021, they are then filtered to only retain those that last more
than 2 days and travel more than 1000 km. This further ensures that only mobile and synoptic-scale cyclones are focused upon,
but means that smaller meso-scale cyclones such as Polar Lows, are not identified. Arctic cyclone tracks are then identified by
those that travel north of Arctic Circle (66.3°N) at any point during their lifetime. This regional filtering matches the domain of
the Berkman et al. (2020a) Arctic shipping database. To assess the hazardous weather conditions that may impact ships travel
on the Earth's surface, the maximum full resolution ERA-5 10-metre wind speed and significant wave height (including tide
and swell) within a 5° radius of the cyclone centre are then identified and added to the cyclone tracks.

## 2.3  Identifying to Intense Cyclones that Caused Rough Sea Conditions in the Arctic

Ships are typically built to withstand moderate intense weather conditions, and only intense Arctic cyclones will pose a
significant threat to ships operating in the Arctic. The Beaufort Wind Scale and Douglas Sea State Scale can be used to gauge
the severity of hazardous conditions over the ocean that could threaten a ship (Simpson, 1906; Schule, 1966; Met Office,
2010). These scales indicate when rough sea conditions are likely to occur depending on surface wind speed or wave height.
The thresholds that result in rough sea conditions are marked as 17 ms$^{-1}$ for surface wind speeds (Beaufort Wind Scale 8 and
higher) and 2.5 m for significant wave heights (Douglas Sea State Scale 5 and higher). These thresholds from the Beaufort
Wind and Douglas Sea State scales are used in this study to identify intense Arctic Cyclones.

The identified ERA-5 Arctic cyclone tracks are filtered to obtain the cyclone tracks that cause rough sea conditions in the
Arctic and have 10 m wind speeds and significant wave heights above these thresholds. Although the exceedence of these
intensity thresholds does not guarantee that every ship will be damaged or affected, these thresholds do provide an objective
measure of the cyclone intensity that can be hazardous for ships. The sensitivity of these thresholds is also assessed in this
study, and the number of ships intersected by cyclones with 10 m wind speeds greater than 25 ms$^{-1}$(Beaufort Wind Scale 10
and higher) and significant wave heights greater than 4 m (Douglas Sea State Scale 6 and higher) is also assessed.

## 2.4  Arctic Shipping Data

Berkman et al. (2020a) published Arctic ship location data from AIS transponders between September 2009 and December
2016, over a domain that includes areas north of the Arctic Circle (66.3°N). Such data could provide insights into the risk
posed by Arctic cyclones to shipping. This dataset includes information such as the timestamp of the AIS transmission, the



unique Maritime Mobility Service Identity (MMSI) number of each ship, and the latitude and longitude positions of all ships that travelled within the Arctic domain and between the given time period.

The Berkman et al. (2020a) data is first concatenated for each unique MMSI per year. Ships are mandated to transmit their
location at a high temporal resolution (i.e., minutes) to ensure safety within the network of mobile ships. This high temporal resolution is reduced from minutes to every hour to match the temporal resolution of ERA-5, by searching for the nearest AIS transmitted timestep to each hour per day. To account for ships having multiple tracks within each month, a new track from each ship is determined if there is a break in the AIS transmission of that ship of more than 48 hours. This break in transmission may be due to the ship being docked and the engine, and therefore the AIS transponder being switched off, signifying the end
of the ships current journey and track.

Past Arctic shipping incidents and accidents data are documented by the Arctic Council (Protection of the Arctic Maritime Environment Agency, 2023). This database reports Arctic shipping incidents from 2005 to 2017 including details on where the incident occurred (i.e., longitude and latitude), the time of the incident, the type of accident/incident. It is also indicated whether the vessel was completely lost or only partially damaged following the incident. Incident south of the Arctic Circle
are included, so this dataset is filtered to retain shipping incidents that occurred north of 66.3°N between September 2009 to December 2016, to match the temporal scale of the Berkman et al. (2020a) ship track dataset. Shipping incidents with no location data are also omitted.

## 2.5   Intersecting Past Arctic Ships Tracks with Intense Cyclone Tracks

The number of past ship tracks intersected by an intense cyclone is quantified to determine the number of ship tracks impacted
by past cyclones. Maximum wind speeds do not occur at the centre of a cyclone, but often occur in the southern quadrant of a cyclone due to near-surface air streams (Browning, 2004; Vessey et al., 2022). Vessey et al. (2022) showed that in the composite structure of the 100 most intense winter and summer Arctic cyclones, the maximum 10-metre wind speeds within these cyclones occur in an area 5° south from the cyclone centre relative to the direction of propagation (see Vessey et al., 2022's Figure S3). Therefore, it would be more representative to use the position of the maximum 10-metre wind speeds and
maximum significant wave heights within the radius of the cyclone to intersect with past ship tracks, to indicate how many ships are impacted by intense weather conditions caused by Arctic cyclones.

In this study, an intersection between a ship track and a cyclone occurs if the ship track is within 3° radius of the maximum 10 m wind (greater than 17 ms$^{-1}$) or significant wave height (greater than 2.5 m). To ensure that this 10 m wind and significant wave height maximum is a caused by the cyclone, it must occur within a 5° radius of the cyclone's 850 hPa relative vorticity
centre. The sensitivity of the number of ships intersected by past cyclones to this distance threshold is also tested, by setting the threshold to 1° and 5° from the point of highest 10 m wind speed and significant wave height.

These intersections between ship tracks and cyclones are then related to shipping incident reports, to determine how many of these ship and cyclone intersections resulted in a reported shipping incident. If a shipping incident was reported within 3° radius of the maximum 10 m wind (greater than 17 ms$^{-1}$) or significant wave height (greater than 2.5 m) of a passing cyclone,
it is counted as an intersection that resulted in a shipping incident. To account for a time lag between an intersection occurring



and the incident being reported, reports within the next 48 hours after the intersection between ship track and cyclone are included.

Ships located on the boundary of the Arctic Circle could be impacted by cyclones with their centre outside and south of this regional boundary. Such cyclones could have their 850 hPa relative vorticity centre south of the Arctic Circle, but still
cause hazardous rough sea conditions within the Arctic Circle. Therefore, cyclone tracks identified from ERA-5 are filtered to retain those that caused 10 m wind speeds and significant wave heights greater than these thresholds within the Arctic Circle $(66.3^{-1}N)$. When a ship track is intersected by an intense Arctic cyclone, this ship track is then excluded from intersecting with the same cyclone at a later timestep. Thus, an intersection between ship track and cyclone is not double counted.

## 3    Trends in Arctic Shipping

Between 2010 and 2016, 176,961 ships with a unique identification number (MMSI) to travel north of the Arctic Circle (Figure 1). The number of ships to travel in the Arctic has increased year-on-year from 2010 to 2016 (Figure 1). This is similarly showed by Berkman et al. (2020b, 2022). In 2010, 15,666 ships with a unique MMSI transmitted an AIS location in the Arctic, whereas this number was more than two times greater in 2016 at approximately 34,780 (Figure 1). This shows that the number of ships operating in the Arctic has increased between 2010 and 2016.

The number of Arctic ships per month varies seasonally, with changes in Arctic sea ice extent, which is also shown by Berkman et al. (2020b, 2022). The maximum number of ships in the Arctic generally occurs in the late summer and early autumn months when Arctic sea ice extent is typically at its annual minimum (Figure 1). The minimum number of Arctic ships generally occurs in winter months (Figure 1). Arctic sea ice extent typically reaches its minimum extent in September and its maximum extent in March (National Snow & Ice Data Centre, 2023). For example, in 2012, Arctic sea ice extent was 15.2
million $km^2$ in March, but had reduced to 3.6 million $km^2$ in September (National Snow & Ice Data Centre, 2023).

The number of ships operating in the Arctic per year has increased by more than double between 2010 and 2016 (Figure 1). However, there is some evidence to suggest that the increasing number of ships in the Arctic has slowed between 2017 and 2019 (NOAA, 2022). Although using a different data source, NOAA (2022) showed that the maximum number of ships per month travelling in the Arctic in 2016 and 2018 were similar, with a maximum of approximately 4,000 ships travelling in the
Arctic in late summer in both years. So the increase in the number of ships operating in the Arctic between 2010 and 2016 (Figure 1) may have slowed from 2016 to 2018. However, given the lack of up-to-date publicly available ship track data, there is insufficient evidence to describe shipping behaviour up to the present day.

## 4    Seasonality in Arctic Ship Tracks

The number of ship tracks in the Arctic varies per month and is highest in summer months when Arctic sea ice is at its
minimum extent. In all winter (DJF) and spring (MAM) months between 2010 and 2016, there were a total of approximately 50,000 Arctic ship tracks (Figure 1). But in all summer (JJA) and autumn (SON) months between 2010 and 2016, there were





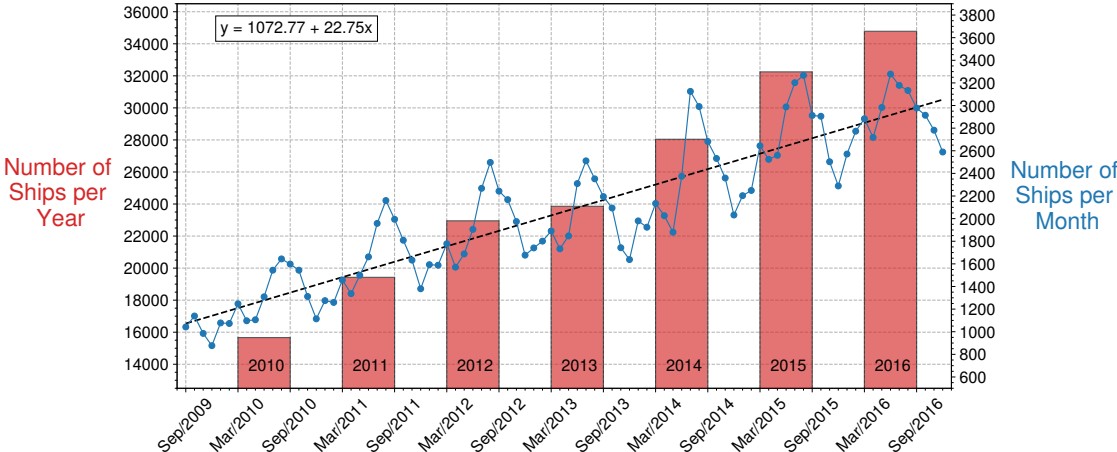

**Figure 1.** The total number of ships with a unique identification number (MMSI) to travel north of the Arctic Circle (66.3°N) per year and month between September 2009 and December 2016 from the Berkman et al. (2020a) Arctic shipping dataset.

more than double the number of Arctic ship tracks than in winter and spring months, with 103,875 and 131,639 Arctic ship tracks occurring in all summer and autumn months respectively (Figure 1).

The highest density of Arctic ship tracks in all seasons occurs in the Barents Sea, and just north of northern Norway in every

season, with more than 200 ships travelling over this region per month in every season (Figure 2e - f). Other regions of high Arctic ship track density occur around Iceland and over Baffin Bay, which is west of Greenland (Figure 2e - f). These regions also have the highest ship track density on annual time-scales (Figure S4 in the Supplementary Material). Ship track density in these regions is greater than 50 ships per month in all seasons. This is similar to Eguiluz et al. (2016), which showed that these areas were the busiest shipping and fishing areas in the Arctic between 2010 and 2014.

There is also a seasonal variation in the spatial distribution of Arctic ship tracks, which shows that in winter (DJF) and spring (MAM), shipping is confined to the Greenland, Norwegian and Barents Seas (Figure 2). However, in summer (JJA) and autumn (SON), shipping is more widespread across the Arctic and there are many more ships travelling across the trans-Arctic shipping routes of the Northern Sea Route (NSR) (along the coastline of Eurasia), the North-West Passage (NWP) (through the Canadian Archipelago). Although, when considering the density of Arctic ship tracks, the number of ships in summer and

autumn is much greater in the Barents Sea than in these trans-Arctic shipping routes (Figure 2e - f).

This spatial distribution in Arctic ship tracks is consistent with Arctic ship tracks described by Corbett et al. (2010); Eguiluz et al. (2016); Hreinsson (2020). So, despite large reductions in Arctic sea ice extent since 1979, trans-Arctic shipping along the NSR and the NWP was limited to summer and autumn, when Arctic sea ice was at its minimum extent. Moreover, the density of trans-Arctic shipping appears much lower than the density of ships in the Barents Sea (Figure 2).





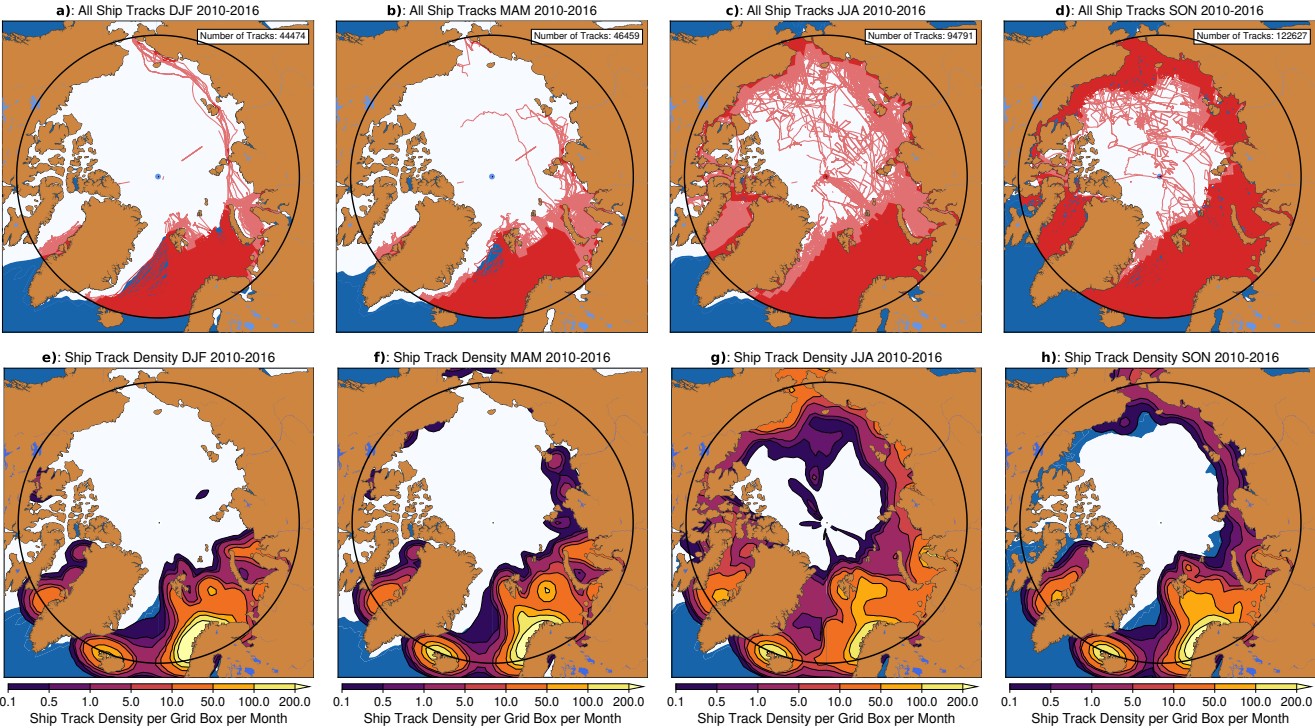

**Figure 2. a) - d)** Ship tracks (red lines) and **e) - f)** ship density per grid box (2.0°N x 5.0°E) per month (**a) and e)** - winter (DJF), **b) and f)** - spring (MAM), **c) and e)** - summer (JJA), and **d) and h)** - autumn (SON)) from September 2009 to December 2016 from the Berkman et al. (2020a) Arctic shipping dataset. Densities are smoothed using a Gaussian filter equal to 1.0. Mean HadISST2.0 Arctic sea ice concentration over each period is shown in white, where sea ice concentration is greater than 15%. The solid black line indicates the Arctic Circle (66.3°N)

## 5  Trans-Arctic Shipping Trends through the Northern Sea Route and North West Passage

One benefit of reduced Arctic sea ice extent is that it offers shorter routes between ports in North America, Europe and Asia. These routes through the Arctic are shorter than traditional mid-latitude routes (see Table S1.1 in the Supplementary Material). The number of ships travelling through the NSR has increased significantly from 2010 to 2016 (Figure 3). In 2010, approximately 90 ships travelled along the NSR between the Kara and Laptev Seas and along the coastline of Eurasia (see red box in Figure 3a), whereas in 2016, approximately 280 travelled through these Seas (Figure 3b). Fewer ships travelled through the NWP (through the Canadian Archipelago) from 2010-2016 than in the NSR (see blue box in Figure 3a). Less than 40 ships travelled in the NWP between 2010 and 2013, but this number was higher in 2014, 2015, and 2016 (Figure 3b). In 2016, approximately 210 ships travelled through the Canadian Archipelago (Figure 3b). However, considering that there are tens of thousands of ship tracks north of the Arctic Circle each year between 2010 and 2016 (Figure 2), the hundreds of ships travelling across the NSR and NWP each year (Figure 3) is significantly less than the thousands of ships travelling in the Greenland, Norwegian and Barents Seas each year.





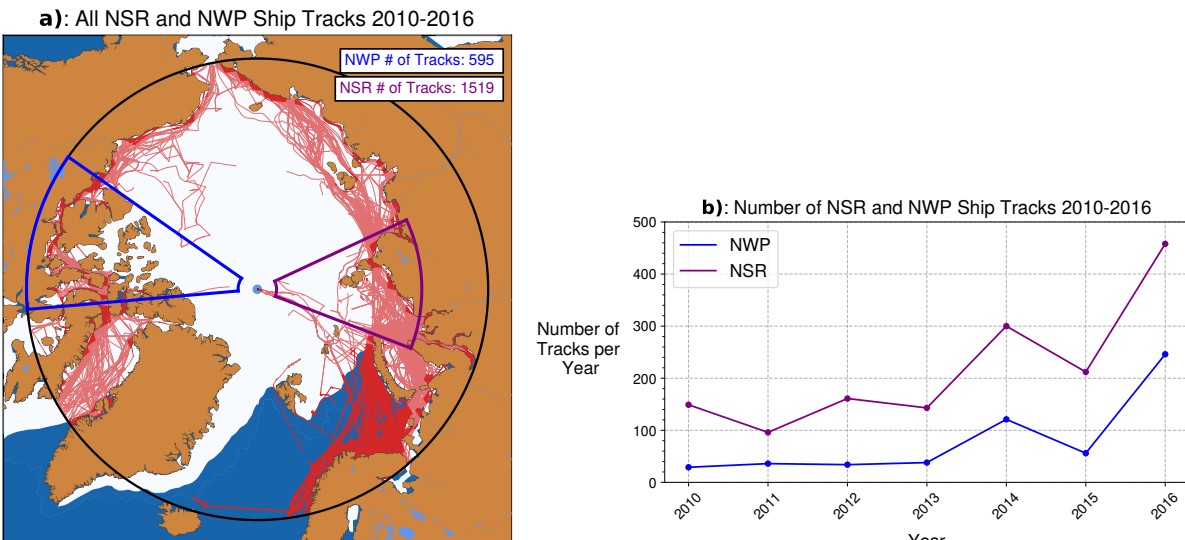

**Figure 3. (a)** All trans-Arctic ship tracks (red lines) that travel through the Northern Sea Route (NSR) (along the coastline of Eurasia) (through the purple box) and the number of ship tracks through the North-West Passage (NWP) (through the Canadian Archipelago) (through the blue box) from January 2010 to December 2016 from the Berkman et al. (2020a) Arctic shipping dataset. The total number of ship tracks across the NSR and NWP are also indicated. Mean HadISST2.0 Arctic sea ice concentration greater than 15% is shown in white. The solid black line indicates the Arctic Circle (66.3°N). **(b)** The annual number of trans-Arctic ship tracks that travel through the NSR (purple line) and NWP (blue line) from January 2010 to December 2016.

Arctic sea ice is much thicker in the Canadian Archipelago and the NWP than sea ice located north of the Eurasian continent and the NSR (Sallila et al., 2019). Therefore, sea ice over the NSR is more susceptible to melting in summer and autumn months and the NSR would have a greater likelihood of being navigable for ships than the NWP. This is a likely reason for the higher
number of ships in the NSR than the NWP between 2010 and 2016 (Figure 3). However, the number of ships travelling through the NSR per year does not show a consistent increase year-on-year, with approximately 300 and 210 ship tracks occurring in 2014 and 2015 respectively (Figure 3b). So, the annual variation in the minimum Arctic sea ice extent may still influence the number of ships travelling across these trans-Arctic shipping routes.

## 6  Number of Past Shipping Incidents

Between 2010 and 2016, there have been a total of 248 reported shipping incidents north of the Arctic Circle (Figure 4a). This is only 0.001% of all ships (176,961) that travelled through north of the Arctic Circle between 2010 and 2016 (Figure 1). These incidents resulted in damage and disruption to the ship and the crew, and include 83 various types of incidents, including capsizing, allision, equipment failure, fire, flooding, grounding, loss of control/propulsion and person overboard (Protection of the Arctic Maritime Environment Agency, 2023).




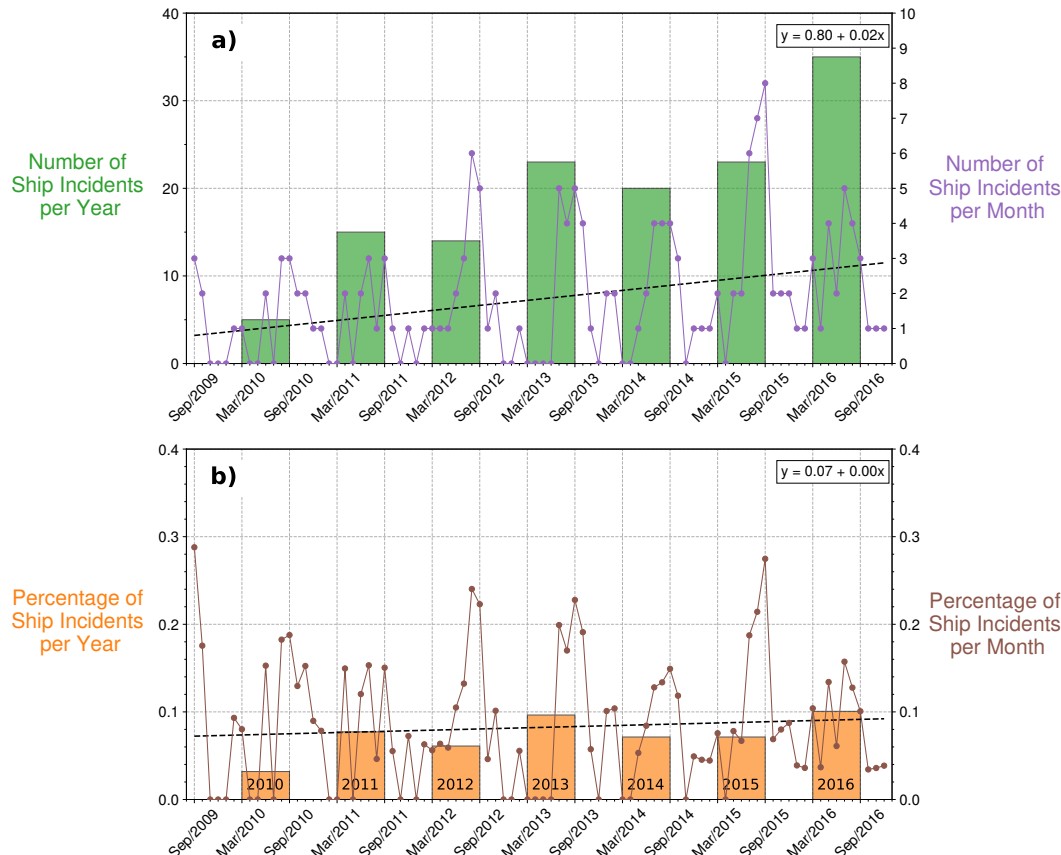

**Figure 4. a)** The total number of ships that reported an incident north of the Arctic Circle (66.3°N) per month and per year between September 2009 and December 2016 from the Protection of the Arctic Maritime Environment Agency (2023) dataset. **b)** The percentage of ships that travelled north of the Arctic Circle (66.3°N) per month and per year between September 2009 and December 2016 (Figure 1 that reported a shipping incident.

For example, the ship with the highest tonnage to report an incident north of th Arctic Circle was the SKF Enisey / SCF Yenisei on $26^{th}$ of September 2014. This ship is a large oil tanker and has a tonnage of 29,844 tn. The report shows that this ship experienced contact with a fixed object in the Kara Sea, consequently leading to marine casualties (Protection of the Arctic Maritime Environment Agency, 2023). The report of this incident aligns with other media reports (e.g., Shipwreck Log, 2014). However, the cause of these incidents is not reported, and it is not indicated if these events occurred due to bad weather such

as Arctic cyclones. This requires matching the position and time information with past Arctic cyclone tracks.



The number of Arctic shipping incidents is generally highest in summer months (Figure 4a), which is when the number of ships operating in the Arctic is generally highest (Figure 1). Up to 35 Arctic shipping incidents were reported per year, with less than 10 Arctic shipping incidents being reported per month between 2010 and 2016 (Figure 1). Only 4 and 35 Arctic shipping incidents were reported in 2010 and 2016 respectively (Figure 4a). Although there is an upwards trend in the number

of reported shipping incidents in the Arctic, these numbers are far less than the total number of ships travelling in the Arctic per year, which in 2016 was approximately 34,000 ships (Figure 1). This shows that less than 0.01% of ships travelling in the Arctic per year reported an incident from any cause. When compared to the total number of ships operating in the Arctic per year (Figure 1), the percentage of ships that reported an incident has remained fairly constant, with approximately 0.003% and 0.01% of all ships per year reporting an incident in 2010 and 2016 respectively (Figure 4b).

**7   Frequency of Intense Arctic Cyclones**

The Beaufort Wind Scale and Douglas Sea State Scale can be used to approximately indicate the intensity threshold of cyclones required to cause rough sea conditions, which may be hazardous to ships (Simpson, 1906; Schule, 1966; Met Office, 2010). Within a $3°$ radius of past Arctic cyclones, the ERA-5 reanalysis dataset suggests that 10 m wind speeds can reach 34 ms$^{-1}$, and significant wave heights including tide and swell can reach 13 m (see Figure 5). These surpass the Beaufort Wind Scale

and Douglas Sea State Scale rough sea thresholds of 17 ms$^{-1}$ and 2.5 m respectively, approximately indicated that cyclones may be hazardous for ships. Generally, winter (DJF) Arctic cyclones are more intense than summer (JJA) Arctic cyclones (see Figure 5), which is in agreement with Zhang et al. (2004); Sorteberg and Walsh (2008); Simmonds et al. (2008); Vessey et al. (2020, 2022). Though, in this study, this is shown in terms of 10 m wind speeds and significant wave heights.

The highest track density of Arctic cyclones with 10 m wind speeds and significant wave heights greater than 17 ms$^{-1}$ and

2.5 m occurs in all seasons over the Barents Sea and around Iceland (Figures 6 and 7). In summer, the track density per season of intense Arctic cyclones appears more extended from the Barents Sea to over the Kara Sea (Figures 6 and 7), perhaps due to the difference in the spatial distribution of summer Arctic cyclones which is highest over the Eurasian coastline (Reed and Kunkel, 1960; Serreze et al., 2001; Simmonds et al., 2008; Crawford and Serreze, 2016; Vessey et al., 2020).

The sensitivity of using a 10 m wind speed threshold of 17 ms$^{-1}$ (Beaufort Scale 8) and significant wave height threshold

of 2.5 m (Douglas Sea State 5) to Arctic cyclone track density is assessed in Figure S3 and S4 in the supplementary material, using 10 m wind thresholds of 25.0 ms$^{-1}$ (Beaufort Wind Scale 10 and higher) and significant wave heights thresholds of 4.0 m (Douglas Sea State 6 and higher). Overall, the spatial distribution shown in the track density of these very intense Arctic cyclones using these more severe intensity thresholds is very similar to that shown in Figures 6 and 7. The highest track density of intense Arctic cyclones in all seasons also occurs over the Barents Sea (Figure S3 and S4 in the supplementary material). But

in summer, the number of Arctic cyclones that exceed these intensity thresholds is much less than all other seasons, and is near zero for summer cyclones with 10 m wind speeds greater than 25.0 ms$^{-1}$ (Figure S3 and S4 in the supplementary material). So, there is very high exposure of ships to intense Arctic cyclones, as the highest density of intense Arctic cyclones occurs where there is the highest density of ships (Figures 2, 6 and 7).





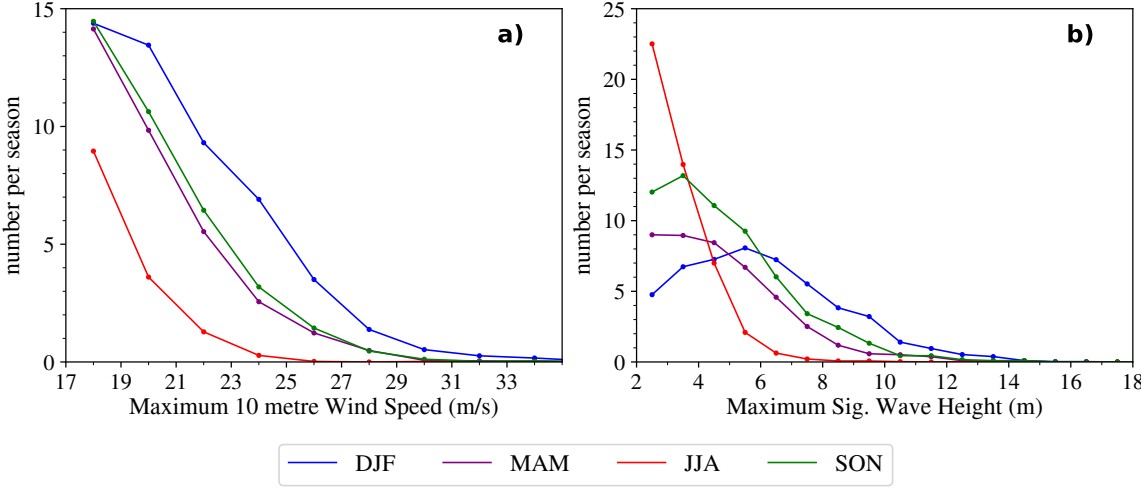

**Figure 5.** The distribution of maximum **a)** 10 metre (m) wind speed and b) significant wave height of Arctic cyclones when they are located in the Arctic. Based on the ERA-5 reanalysis dataset between 1979-2021 in spring (MAM), summer (JJA) and autumn (SON), and 1979/80-2020/21 in winter (DJF). Bin widths are **a)** 2 ms$^{-1}$ for 10 m wind speed and **b)** 1 m for significant wave height. Note the difference in y-axis scales in **a)** and **b)**.

## 8 Number of Ships Intersected by Past Intense Arctic Cyclones

Between 2010 and 2016, there have been tens of thousands of ship tracks per year located and intersected within the vicinity of Arctic cyclones strong enough to cause rough sea conditions (Figure 8). For example, 246,690 ship tracks were located within 3° and intersected by an Arctic cyclones' maximum significant wave heights greater than 2.5 m (Figure 8b). But only 18 of these 246,690 ship track and cyclone intersections (less than 0.0001%) coincided with a reported shipping incident (Figure 8d), suggesting that most Arctic ship tracks are unaffected by the passage of an intense Arctic cyclone.

It is also noteworthy that the number of ship tracks intersected by past intense Arctic cyclones is a high percentage of all ship tracks. Of the 334,944 ship tracks in the Arctic between 2010 and 2016 (Figure 2, 73% of these were intersected by an intense Arctic cyclone. Therefore, it is actually very common for a ship travelling in the Arctic to be impacted by an intense Arctic cyclone on its transit. yet there are very few cases where this led to a reported shipping incident.

The number of ship tracks intersected by past Arctic cyclones with significant wave heights greater than 2.5 m is much
greater than the number of ship tracks intersected by past Arctic cyclones with 10 m wind wind speeds greater than 17 ms$^{-1}$. In 2016, approximately 10,000 ship tracks were located within 3° area of wind speeds greater than 17 ms$^{-1}$ (Figure 8a), but approximately 60,000 ship tracks within areas where waves were greater than 2.5 m (Figure 8b). This suggests that extreme waves caused by Arctic cyclones occur more generally where Arctic ship density is highest, i.e., over the Barents Sea, perhaps due to large fetch of waves over the North Atlantic Ocean.



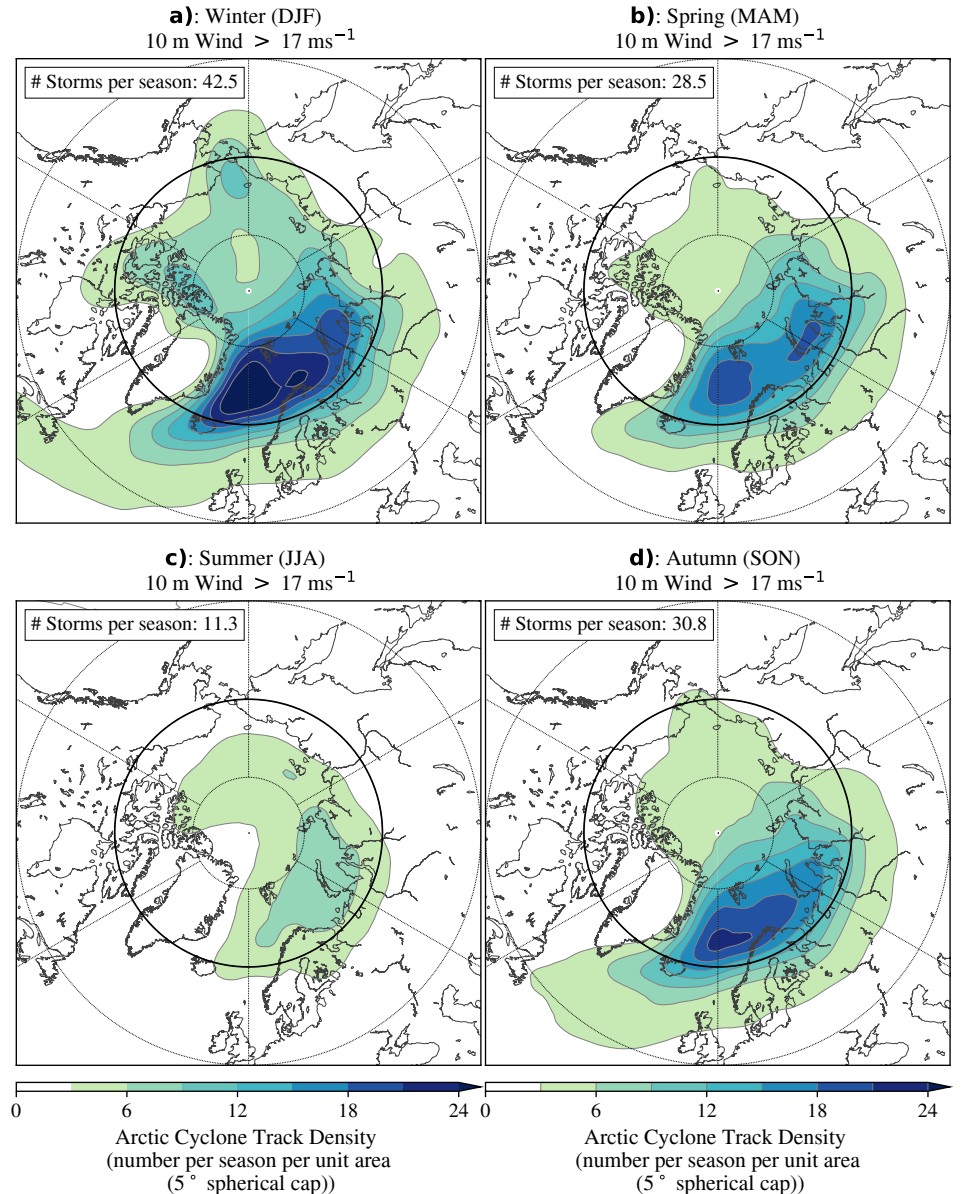

**Figure 6.** Track density per season of **a)** winter (DJF), **b)** spring (MAM), **c)** summer (JJA) and **d)** autumn (SON) cyclones that have maximum 10 metre (m) wind speeds in the Arctic (66.3°N) greater than 17 ms$^{-1}$. Determined from the ERA-5 reanalysis dataset between 1979-2021 in spring (MAM), summer (JJA) and autumn (SON), and 1979/80-2020/21 in winter (DJF). Track density indicates the number of cyclones that travel over a grid point and has units of number per season per unit area (5° spherical cap, approximately 10$^6$ km$^2$). Longitudes are shown every 60°E, and latitudes are shown at 80°N, 66.3°N (bold) and 50°.



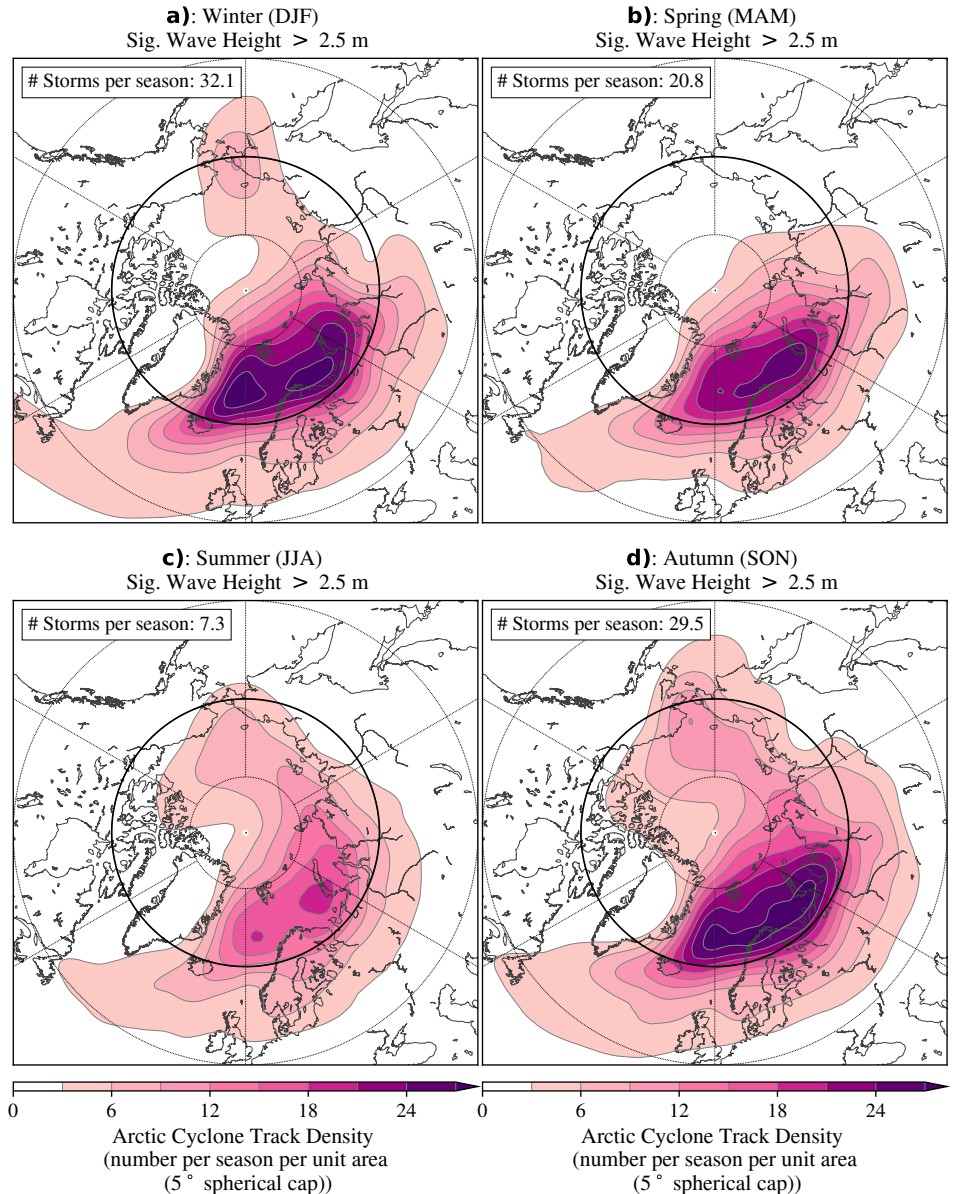

**Figure 7.** Track density per season of **a)** winter (DJF), **b)** spring (MAM), **c)** summer (JJA) and **d)** autumn (SON) cyclones that have maximum significant wave height including tide and swell in the Arctic (66.3°N) greater than 2.5 m. Determined from the ERA-5 reanalysis dataset between 1979-2021 in spring (MAM), summer (JJA) and autumn (SON), and 1979/80-2020/21 in winter (DJF). Track density indicates the number of cyclones that travel over a grid point and has units of number per season per unit area (5° spherical cap, ≈ $10^6$ km$^2$). Longitudes are shown every 60°E, and latitudes are shown at 80°N, 66.3°N (bold) and 50°.





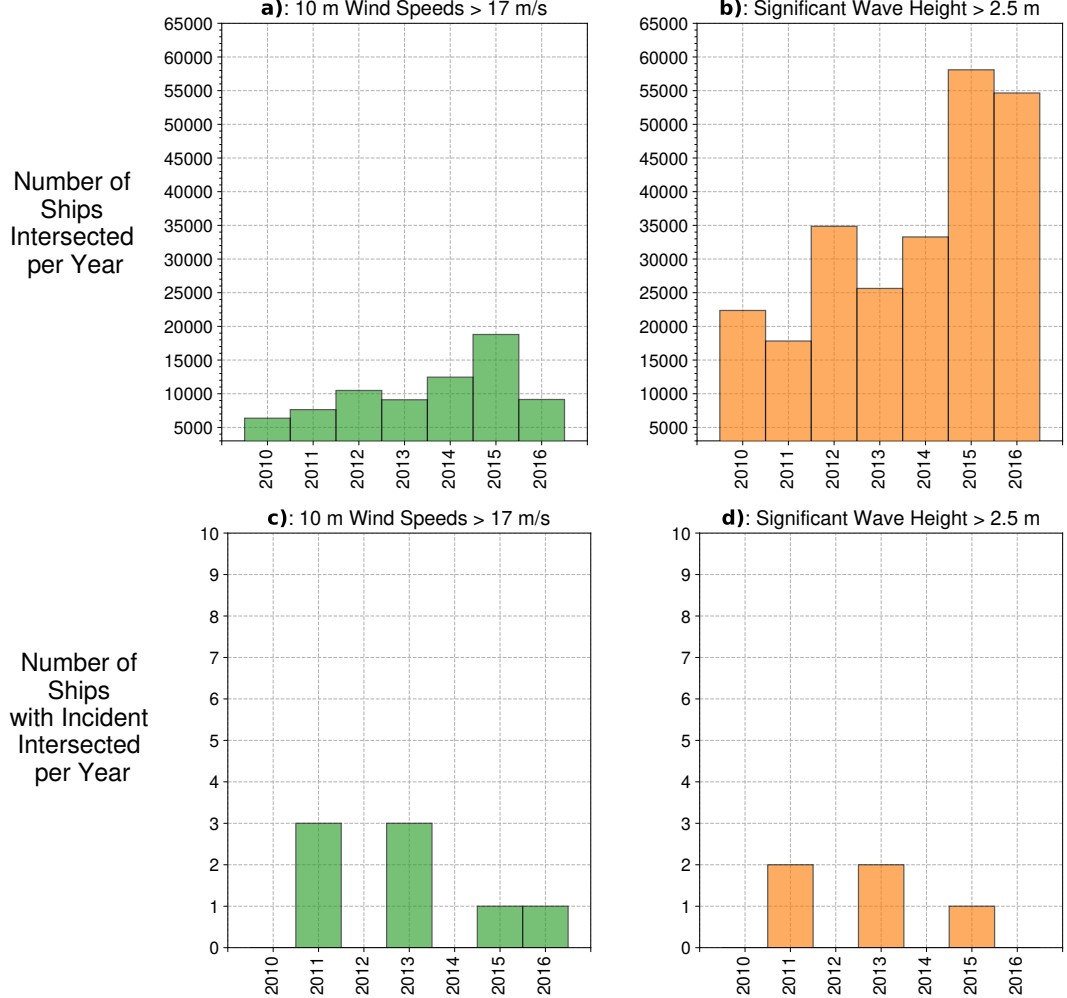

**Figure 8.** The annual number of intersections between ship tracks north of the Arctic Circle (66.3°N) within a 3° radius of an Arctic cyclones' (any cyclone to travel north of 66.3°N) maximum **a)** 10 metre wind speeds greater than 17 ms$^{-1}$ and **b)** significant wave height including tide and swell greater than 2.5 m, between January 2010 and December 2016. **c)** and **d)** show the number of shipping incidents north of the Arctic Circle (66.3°N) that were reported within 48 hours of a ship track and cyclone intersection. Note: multiple intersections between a cyclone and the same ship track are not double counted.

The number of ship tracks in the vicinity of and therefore intersected by past intense Arctic cyclones with significant wave heights greater than 2.5 m has more than doubled from 2010 to 2016, from approximately 20,000 intersections in 2010 and 2011 to approximately 55,000 intersections in 2015 and 2016 (Figure 8b). This increasing trend in the number of ship tracks intersected by past intense Arctic cyclones is most-likely due to the year-on-year increase in the number of ship tracks operating in the Arctic (Figure 1). But despite this increasing number of intersections between ship tracks and Arctic cyclones that cause





rough sea conditions, the number of reported shipping incidents caused by the passage of Arctic cyclone remains extremely small, and has not increased from 2010 to 2016.

Between 2010 and 2016, only a total of 8 (Figure 8c) and 5 (Figure 8d) Arctic shipping incidents were reported within 48 hours of an intersection between a ship track and 3° radius of an Arctic cyclones' maximum 10 m wind speeds greater than 17 ms$^{-1}$ or maximum significant wave height greater than 2.5 ms$^{-1}$ respectively. However, all of these only resulted in partial

damage to the ship, and none of these intersections resulted in the ship being lost (Tables S2 and S3 in the Supplementary material). Even when the distance from the Arctic cyclones' maximum 10 m wind speeds greater than 17 ms$^{-1}$ or maximum significant wave height greater than 2.5 ms$^{-1}$ is extended to 5.0°, only 10 and 7 Arctic shipping incidents were reported following the passage of an Arctic cyclone (Figure S8 in the Supplementary Material). These shipping incidents include fire, allision/collision, equipment failure, loss of electrical power and loss of control (Tables S2 and S3 in the Supplementary

material). The consequences of these incidents included marine casualties (Table S2 and S3 in the Supplementary material). So of the 248 reported Arctic shipping incidents between 2010 and 2016, only very small percentage of these incidents (less than 0.01%) were preceded by a passage of Arctic cyclone that caused rough sea conditions.

It is surprising how frequently Arctic ship tracks are intersected by an intense Arctic cyclone, as the track of an intense Arctic cyclone would likely be communicated through weather forecasts. Given that ships likely have radio or weather forecast

equipment onboard, the ship would likely be aware of the incoming intense cyclone. A publicised example of this is the SS El Faro shipping disaster in 2015, where the ship knowingly (from weather forecasts) travelled into the track of Hurricane Joaquin, which led to the sinking of that ship (Fedele et al., 2017; Vanity Fair, 2018). It would be expected that ships may avoid the forecasted paths of the intense Arctic cyclones. As there is are a very large number of intersections between ship tracks and intense Arctic cyclones, this does not appear to be the case (Figure 8). Perhaps ships are able to withstand and continue to

operate in such hazardous conditions caused by intense Arctic cyclones.

## 9  How do Ships Respond to the Passage of an Intense Arctic Cyclone?

To gauge how ships respond to the passage of intense Arctic cyclones, the most intense cyclones between 2010 and 2016 that travel through the busiest cluster of Arctic ships (in the Barents Sea - Figure 2), have been identified within ERA-5. According to ERA-5, the five most intense Arctic cyclones that caused the highest and most intense significant wave heights in the Barents

Sea between 2009 and 2016 occurred in Dec. 2012, Dec. 2014, Feb. 2015, Mar. 2015 and Mar. 2016, where significant wave heights in the Barents Sea exceeded 10 m (Figures 9 and 10).

It would be expected that ships may avoid the paths of these most intense Arctic cyclones. This however does not seem to be the case. Ships are located within the regions of the highest 10 m wind speeds and tallest waves (Figures 9 and 10). Ships appear able to withstand and even travel through the passage of the most intense historical Arctic cyclones. Furthermore, when

using information from the Protection of the Arctic Maritime Environment Agency (2023) database, none of these most intense Arctic cyclones resulted in a reported shipping incident (Table S2 and S3 in the Supplementary Material).



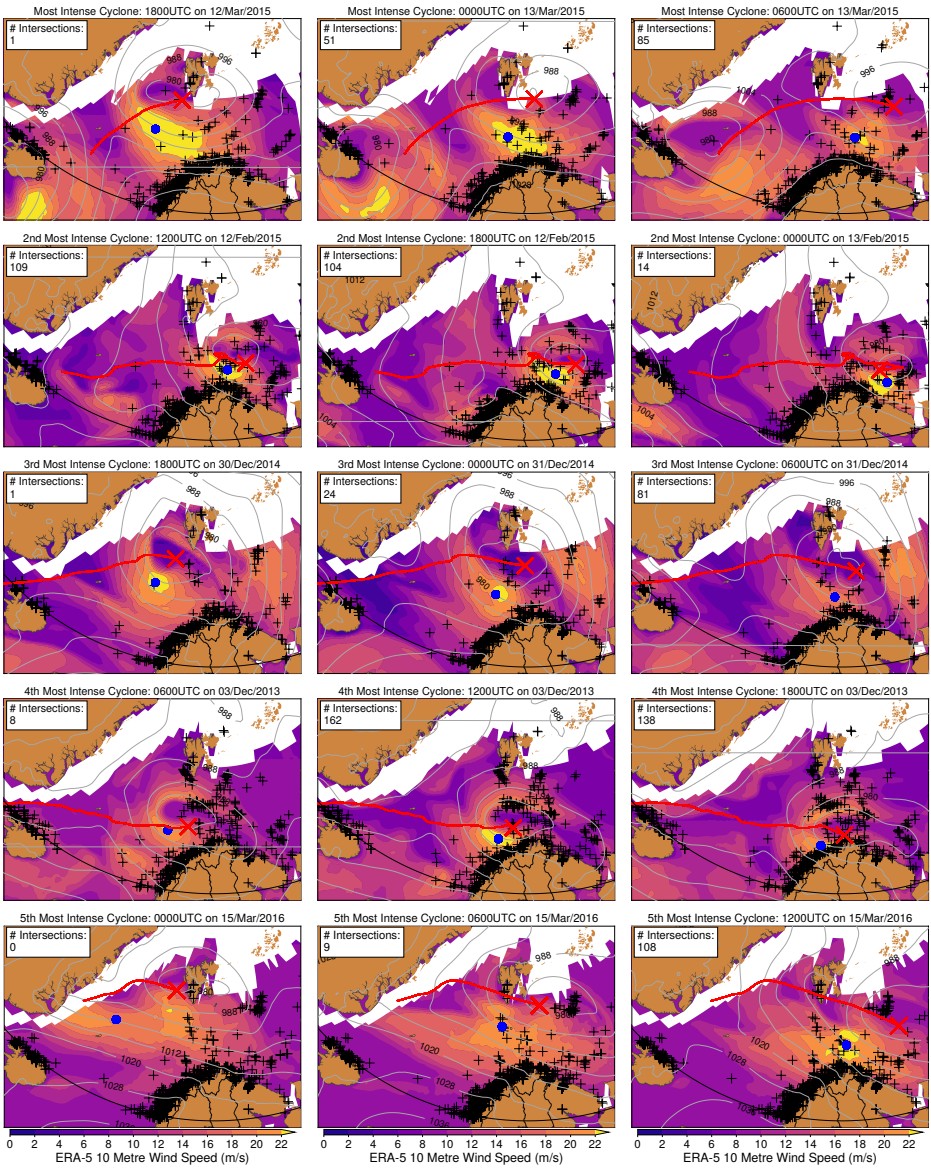

**Figure 9.** The tracks of the most intense (by significant wave height including tide and swell) Arctic cyclones to occur over the Barents Sea (between 20-30°E and 71-77°N) from 2009 to 2016 according to ERA-5. Each row shows the track of each cyclone from the most intense to the $5^{th}$ most intense. The cyclone 850 hPa relative vorticity centre is denoted by the red cross, and its track by the red line. The location of the maximum significant wave height within a 5° radius of the cyclone centre is denoted by the blue marker, with the number of ships intersecting within 3° of this location also given. 10 metre wind speed is given by the contoured colours, and the sea level pressure is given by the grey contours. Arctic sea ice extent from the HadISST2.0, where sea ice concentration is >15% is indicated in white. The black crosses denote the AIS satellite positions of ships operating in the Arctic at this time.



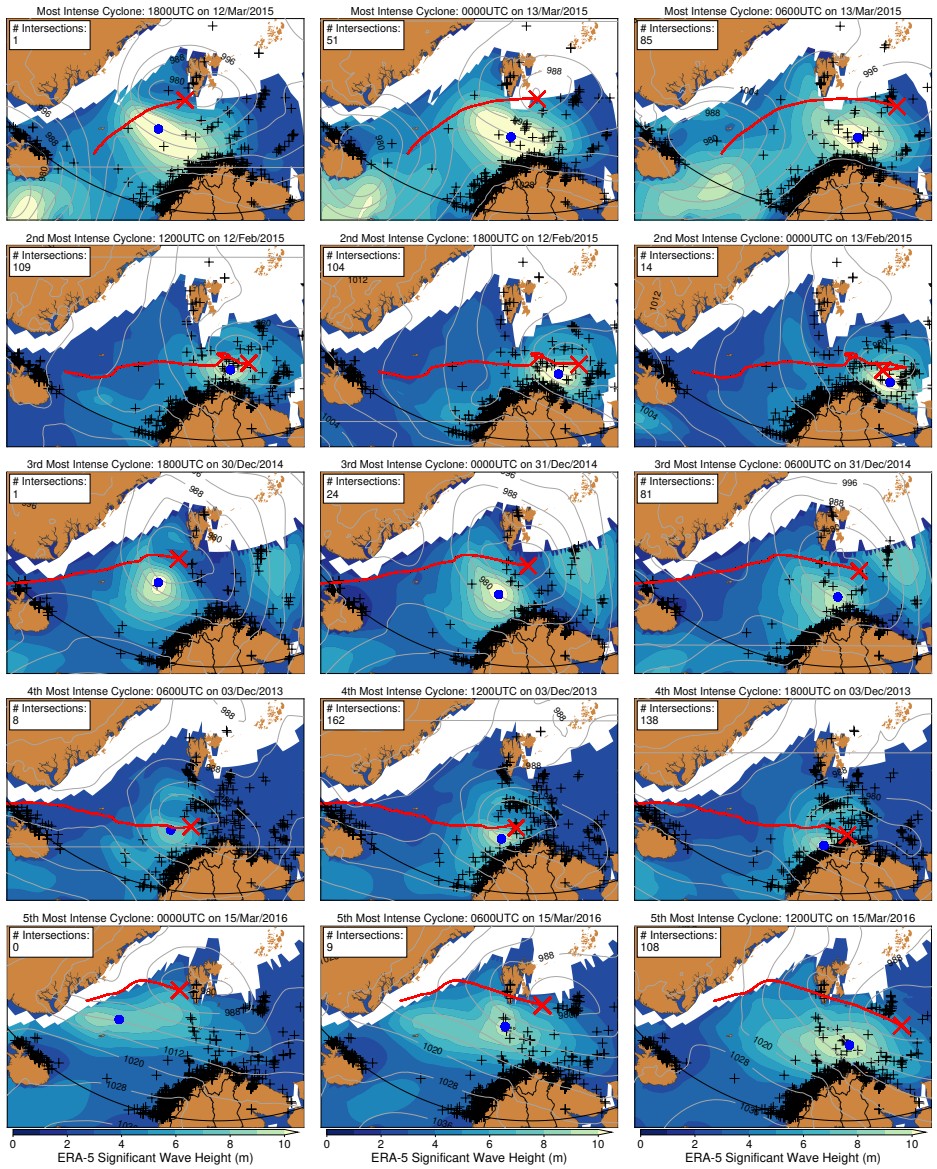

**Figure 10.** The tracks of the most intense (by significant wave height including tide and swell) Arctic cyclones to occur over the Barents Sea (between 20-30°E and 71-77°N) from 2009 to 2010 according to ERA-5. Each row shows the track of each cyclone from the most intense to the $5^{th}$ most intense. The cyclone 850 hPa relative vorticity centre is denoted by the red cross, and its track by the red line. The location of the maximum significant wave height within a 5° radius of the cyclone centre is denoted by the blue marker, with the number of ships intersecting within 3° of this location also given. Significant wave height is given by the contoured colours, and the sea level pressure is given by the grey contours. Arctic sea ice extent from the HadISST2.0, where sea ice concentration is >15% is indicated in white. The black crosses denote the AIS satellite positions of ships operating in the Arctic at this time.



So perhaps the risk of total loss of the ship and its cargo is low and mitigated by the ships ability to withstand the most intense conditions. However, other than direct damage to the ships and its cargo, the ship could experience business interruption due to the passage of intense Arctic cyclones. This could lead to a delay in the ships transit, if the ship has to slow down and prepare for the cyclones approach. Other damage could occur to port facilities, which is not considered here. So despite the most intense cyclones not being hazardous to ships in the Arctic and likely not causing direct damage, other indirect damage could occur leading to a financial loss.

## 10   Conclusions

The risk posed by Arctic cyclones to ships has seldom been quantified due to the lack of publicly available data of past Arctic ship tracks. Such data is instead often privately owned and difficult and costly to obtain. However, the lack of publicly available historic ship track data has been somewhat reduced by Berkman et al. (2020a), who made Arctic shipping data publicly available for a limited time period between September 2009 and December 2016. Such a dataset can be used to gain new insights into Arctic shipping behaviour and investigate the risk of Arctic cyclones to shipping.

This study explores annual trends and seasonality in the number of ships travelling north of the Arctic Circle (66.3°N), and Arctic shipping incidents. Historic ship tracks are combined with the tracks of past intense Arctic cyclones, to determine the number of ships intersected by intense Arctic cyclones, and the number of these that coincided with a reported shipping incident. Overall, the number of ships in the Arctic exposed to intense Arctic cyclones is increasing. This is primarily driven by an increasing number of ships operating in the Arctic, which may be related to decreasing Arctic sea ice extent that is a consequence of global warming. Intense Arctic cyclones are found to very frequently intersect with Arctic ship tracks, with tens of thousands of intersections occurring each year. But the percentage of intersections that lead to a reported shipping incident is extremely small, and is less than 0.0001%. This suggest that past Arctic cyclones are not hazardous to ships. Instead, ships appear to be able to travel into and withstand the most hazardous conditions caused by the most hazardous Arctic cyclones.

   – The number of ships in the Arctic has increased year-on-year between 2010 and 2016, but there number of shipping incidents has remained fairly stable between 2010 and 2016

It is expected that the significant decrease in Arctic sea ice extent over the last few decades in response to global warming has led to more shipping in the Arctic (e.g., Browse et al., 2013; Lasserre, 2014; Melia et al., 2016; Lasserre, 2019). It is shown that the annual number of ships operating in the Arctic has in fact increased year-on-year from 2010 to 2016, similarly to Berkman et al. (2020b, 2022). In 2010, a total of 15,666 unique ships travelled north of the Arctic Circle, but this number was more than double in 2016, where 34,780 unique ships travelled in the Arctic. The maximum number of ships operating in the Arctic appears to coincide with the minimum Arctic sea ice extent in September. However, despite this doubling in the number of ships operating in the Arctic, the number of reported shipping incidents has remained fairly stable from 2010 to 2016, with between 300 and 500 reported shipping incidents occurring per year. In fact, as a percentage of the total number of ships, the number reported shipping incidents from all causes, including those unrelated to the passage of a cyclone, has actually decreased by half from 2% in 2010 to 1% 2016.



– The spatial density of Arctic ships is greatest over the Barents Sea year-round, and the number of ship tracks along the Northern Sea Route (along the coastline of Eurasia) and North-West Passage (through the Canadian Archipelago) is much greater in summer (JJA) and autumn (SON) than in winter (DJF) and spring (MAM), when Arctic sea ice extent has sufficiently retreated enough from the coastline

Arctic ship track density is greatest year-round in the Barents, Greenland and Iceland Seas. This is especially the case in
winter and spring, where Arctic ships are rarely found outside of this region. In summer and autumn, Arctic ship tracks are found across most of the southern Arctic, but still highest over the Barents, Greenland and Iceland Seas, consistent with Corbett et al. (2010) and Eguiluz et al. (2016). In summer and autumn, Arctic shipping also occurs over Baffin Bay (near Greenland) and along the shipping routes of the Northern Sea Route (NSR) (along the coastline of Eurasia) and North-West Passage (NWP) (through the Canadian Archipelago). The number of ships travelling through the NSR and NWP has increased from 2010 to
2016. Of these shipping routes, the NSR typically has a higher number of ship transits than the NWP. This is likely due to the Northern Sea Route being more typically ice-free in summer and autumn months than the North-West Passage, where sea ice tends to be thicker and less susceptible to melting.

    – Shipping incidents do occur in the Arctic, but the vast majority of incidents do not occur following the passage of an intense Arctic cyclone

Past Arctic ship tracks have been combined with Arctic cyclone tracks derived from ERA-5 to determine the number of ships intersected by intense Arctic cyclones, and how ships respond to the passage of intense Arctic cyclones. Shipping incidents published by Protection of the Arctic Maritime Environment Agency (2023) are also used to confirm whether these intersections between ship tracks and cyclones resulted in a shipping incident. The track density of intense Arctic cyclones that cause rough sea conditions is greatest over the Barents, Greenland and Iceland Seas in all seasons, with the frequency typically being
highest in winter.

Between 2010 and 2016, there were 248 reported Arctic shipping incidents. This is only 0.001% of the total number of ships that travelled north of the Arctic Circle, as there were 176,961 unique ships that travelled in the Arctic Circle between 2010 and 2016. Thus, shipping incidents did occur in the Arctic between 2010 and 2016, but they are rare. Of these reported shipping incidents, only approximately 2% are found to occur following the passage of an intense Arctic cyclone. So out of the shipping
incidents that do occur, the vast majority appear unrelated to the passage of intense Arctic cyclones.

Of all of the Arctic ship tracks between 2010 and 2016, approximately 66% were intersected by an intense Arctic cyclone, that caused rough sea conditions. Such a high percentage is surprising given that the ships would likely have prior knowledge of the passage of an intense Arctic cyclone from weather forecasts, and may avoid the path of an intense cyclone. But less than 0.0001% of these intersections between intense Arctic cyclones and ship tracks resulted in a reported shipping incident. So,
Arctic cyclones are found to frequently impact Arctic ship tracks, which did not result in an shipping incident. This suggests that ships may have low vulnerability and are fairly resilient to the rough sea conditions that can be caused by the most intense Arctic cyclones. Therefore, Arctic cyclones may not be hazardous to ships.



So perhaps the risk of synoptic-scale Arctic cyclones to ships and its crew and cargo is low and mitigated by the ships ability to withstand the most intense conditions. However, ships could also experience other consequences than damage to the
ship and crew, such as business interruption and delays in transit. Other damage which is not considered in this study could occur to port facilities. This study is also limited to using ship track data from September 2009 to December 2016. Although we conclude that synoptic scale cyclones pose little hazard to Arctic shipping that is not to say that severe weather is not a problem for shipping in the Arctic. For example, polar lows, which are smaller scale phenomena in the sub-polar seas have been implicated in the loss of a numerous small vessels (Rasmussen, 2003) and often have an impact on the normal shipping
operations (Moreno-Ibáñez et al., 2021). To better understand the risks to shipping, extensive and up-to-date ship track and incident data needs to be more publicly available. Data needs to be up-to-date so that the risks to shipping can be monitored as global warming continues to rapidly change the Arctic.



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

*Competing interests.* The contact author has declared that neither they nor their co-authors have any competing interests.

*Acknowledgements.* The authors acknowledge the funding and support from the Scenario NERC Doctoral Training Partnership Grant

(NE/L002566/1) and co-sponsor, AXA XL, in the development of this research. The authors would also like to acknowledge the European Centre for Medium-Range Weather Forecasts (ECMWF) for the production of ERA5 reanalysis dataset. The work described in this paper has received funding from the European Union's Horizon 2020 Research and Innovation programme through Grant agreement no. 727862 APPLICATE. The content of the article is the sole responsibility of the author(s) and it does not represent the opinion of the European Commission, and the Commission is not responsible for any use that might be made of information contained.