# Peer review of "The Risk of Synoptic-Scale Arctic Cyclones to Shipping"

_EGUsphere, 2023_

## Referee Comment (RC1)

**The Risk of Synoptic-Scale Arctic Cyclones to Shipping**

Alexander F. Vessey, Kevin I. Hodges, Len C. Shaffrey, and Jonathan J. Day

This is an interesting and potentially important paper because of its creative integration with open-access data repositories, addressing transdisciplinary questions with societal relevance, and it should be published as an exemplar.

The integration of Satellite Automatic Identification System (S-AIS) data to interpret incidents with maritime ship traffic in relation to sea-ice distributions and cyclonic weather events is innovative with the intersecting dynamics of biogeophysical and socioeconomic systems north of the Arctic Circle.   The following comments are intended to enhance the publication with appreciation to the authors.

1. The Introduction presents well-reasoned conclusions about Arctic maritime ship traffic as a starting point to formulate the questions that stimulated this research.

2. The Methods includes five subsections that are fine.  However, a cross-cutting subsection should be added to clarify the transdisciplinary integration approach.  More specifically, this additional subsection should provide details about the mapping methodologies and geospatial analyses with Big Data.

3. In Methods subsection 2.1: The application of sea-ice data from the Hadley Centre in the United Kingdom complements previously published applications of sea-ice data from the National Snow and Ice Data Center in the United States. Noting the data integration includes the same S-AIS data in Berkman et al. (2020, 2022), how do ship-ice patterns or trends compare with data from the two sea-ice repositories?

4. In Methods subsection 2.2: *"Arctic cyclones are identified in hourly ERA-5 data."* As a data integration opportunity, it would seem helpful to mention the relative rates of S-AIS (e.g., satellite orbit) and sea-ice (e.g., daily) data production from the relevant repositories in relation to hourly ERA-5 data.

5. Ships and storms have tracks.  It would seem helpful to be consistent with terminology throughout the paper about "ship tracks" and "storm tracks."

6. "The Arctic Circle (66.3ºN) is incorrect."  To be corrected throughout the paper: the Arctic Circle is either 66º33'47.5"N or 66.5ºN?

7. In Methods subsection 2.3: "*The sensitivity of these thresholds is also assessed in this study, and the number of ships intersected by cyclones with 10 m wind speeds greater than 25 ms−1(Beaufort Wind Scale 10 and higher) and significant wave heights greater than 4 m (Douglas Sea State Scale 6 and higher) is also assessed.*" However, the details are in subsection 2.5. "*In this study, an intersection between a ship track and a cyclone occurs if the ship track is within 3◦ radius of the maximum 10 m wind (greater than 17 ms−1) or significant wave height (greater than 2.5 m).*"  Additionally, what is a 3º radius in relation to distance?

8. In Methods subsection 2.5: "The number of past ship tracks intersected by an intense cyclone is quantified to determine the number of ship tracks impacted by past cyclones." How? This is the crux of the matter with the missing subsection that is noted above.

9. Tables and Figures:

   a. With the five or six Methods subsections, it would seem reasonable to create one synthesis figure for each subsection. Currently, there are 10 figures and this number should be reduced in the main paper and any ancillary figures could be part of the Supplement.

   b. Conversely, the Tables about shipping incidents in the Supplement are new and should be included in the main paper.

   c. Reducing the number of figures may require the Methods subsections to be reorganized in parallel to the story that will be woven through the discussion, introducing Arctic Shipping Data earlier, perhaps as subsection 2.1. The logic of the paper would be tightened by such symmetry.

   d. It would seem helpful to generate a single synthesis figure with seasonal (left to right and spell-out in the legend) December-January-February (DJF), March-April-May (MAM), June-July-August (JJA) and September-October-November (SON):
      i. "ship tracks";
      ii. "ship density" (noting "ship tracks" is based on individual ships per time and "ship density" is based on all ships per area per time);
      iii. Arctic cyclone wind speed;
      iv. Arctic cyclone wave height.

   e. In Figure 3: It is interesting that the annual ship-track trends for the Northwest Passage and Northern Sea Route are nearly parallel from 2010-2016, with relative increases in 2014 and 2016. This complements annual ship-ice interactions across latitudes shown in Berkman et al. (2020, 2022).

10. Discussion: A paragraph or two should be added about the specific creative applications in this paper as well as generalized utility of open-access data repositories to empower open-ended inquiry, notably in view of risk analyses that are the focus of the journal.

11. Discussion: Lastly, the audiences for this paper include the decisionmaking institutions that have remits to address maritime ship traffic in relation to the biogeophysical dynamics of the Arctic Ocean. It would seem relevant to mention or specify the binding international agreements that address safety of life at sea, search and rescue, pollution prevention, and marine ecosystem impacts.

---

## Referee Comment (RC2)

**General comments**

Vessey et al. describe recent trends in ship traffic within the Arctic Circle, and measure the risks they face due to severe weather.

The article contains a clear description of the aims of the investigation, its wider context, data and methods used, and explanations of results. It delivers new information on the dangers posed by severe weather to Arctic shipping, of particular benefit to those with exposure in this sector. However, the article has significant shortcomings, and I recommend a major revision before publication.

I ask the authors to address the following weaknesses in a revised version of the manuscript: (i) a significant number of grammatical mistakes, (ii) inconsistencies in the values of quantities quoted in the text (often percentages are wrong by a factor 100, and other data in discussions do not match plotted values in figures), and (iii) the proxy of severe weather (*a ship within 3 degrees of the location of the maximum wind or wave height above a threshold which itself is within five degrees of an analysed cyclone centre*) is unnecessarily complicated, and introduces avoidable error in identifying the true shipping hazard, namely roughness of the sea state at a ship's location.

**Major comments**

**1.** While the general standard is good, there are many mistakes in the grammar. Some of the more significant errors have been listed in the section 'Minor Comments'. The authors are asked to proofread the entire manuscript for grammatical errors, and meet journal standards.

**2.** The pdf file has an unusual size of 180 MB and some Apps struggle to load and display a file of this size. A reduction of file size would be useful for the audience. My guess is that Figures 2a-d might be of huge size, because they contain plots of more than 300,000 unique ship tracks. If this guess is correct, then converting them to a different file format may help, or moving Figures 2a-d to Supplementary Information might be suitable?

**3.** All quoted data values in the text need greater inspection by the authors. There is an unusual number of inconsistencies between data values in the text and values plotted in Figures. Some mismatches have been in the 'Minor Comments' section below, but I encourage the authors to comprehensively review this aspect of their manuscript.

**4.** There is potential to improve the Introduction.

   a) The opening paragraph of the Introduction describes how sea-ice is declining hence the Arctic is becoming more accessible to shipping. The authors may wish to give a more complete description of Arctic changes in the opening paragraph, before focussing on benefits to shipping. For example, add a sentence or two on its negative aspects both locally

(people, wildlife, adapted to colder climate) and on mid-latitude climate – implicated in colder Siberian winter, and droughts/wildfires/floods in mid-latitude summer, e.g. Coumou et al. (2018). Further, the authors may wish to revise their opening statement to reflect how global warming is considered to be the *main* driver of sea-ice decline. (Multidecadal variability in the climate system may play a minor role.)

b) The Introduction contains multiple references to information in Table S1. The fact that data in Table S1 is central to this discussion suggests it is more appropriately placed in the main manuscript, either as a Table, or as a map with routes (with polar stereographic map projection).

c) My understanding is that the analysis of shipping damage data from the Arctic Council is a key contribution, and an extra paragraph on this dataset in the Introduction would help the reader.

**5.** Revision to analysis methods. The authors use the concept of an intersection of Arctic cyclones containing maximum winds/wave heights above a threshold, with the locations of ship tracks, to identify incidents due to severe weather. This contrasts sharply with the simpler method of identifying severe conditions based on wave height along the ship track. ERA5 provides all the data required to identify wave height at all locations of a ship track. Could the authors explain why winds are used together with wave heights in a study of ship damage? Do strong winds cause damage, per se? (I thought it was the sea state that damaged modern vessels, but not an expert.) Further, could the authors explain the benefits of including the identification of cyclones here? The definition of severe weather as a ship within 3 degrees of significant winds or waves seems like a proxy for wave height, but given that wave height is available, why use cyclone tracking?

**Minor comments/corrections**

1. lines 18-19: the sentence beginning 'In September 1980…" provides an indication of the sea-ice decline using September data for 1980 and 2012. I suggest expressing the decline in terms of annual mean Arctic sea-ice over the past 40 years, using data from NSIDC, and expressed in % change (decline) per decade, to create a more robust and representative measure of the decline.

2. line 21: the word 'however' is not appropriate since there is no contrasting information in this sentence.

3. the Arctic Circle is defined throughout as 66.3°N, whereas the standard definition is 66°34'N, or approx. 66.6°N (e.g. https://www.pmel.noaa.gov/arctic-zone/faq.html).

4. line 75: "combined" should be combine?

5. line 80: grammar error in "has been somewhat been addressed"

6. lines 94-96: the description of the sections in the manuscript is not accurate. For example, the summary is given in Section 10, rather than Section 4.

7. line 139: error in grammar

8. line 169: "Incident south of…" should be "Incidents south of…"

9. line 171: "to match the temporal scale" could be replaced by "to match the spatio-temporal scale"

10. line 175: "southern quadrant" – either southern half, or southwest quadrant ('southern' is not a quadrant)

11. line 197: error in representing the degrees symbol (a superscript '-1')

12. lines 201-202: "This is similarly showed by" to "This is similarly shown by"

13. lines 204-205: do the authors have any insight into the extent that the increasing number of observed ships is caused by the increasing use of transponders (lines 67-68)? Is it possible to distinguish all large ships over 300 tonnes, or passenger ships, and identify their trend?

14. Could the scale in Figures 2e-2h be extended to include low values of track density close to the pole?

15. lines 225ff : should references to Figure 2e – f be replaced by Figure 2e – g?

16. line 242: reference to a "Table S1.1" needs corrected.

17. lines 244-248: the numbers in the text do not match with the plotted values in Figure 3b.

18. line 261: "only 0.001% of all ships" – inaccurate: 248 incidents among 176,961 ships = 0.14%.

19. line 273: reference to Figure 1, should be Figure 4a?

20. lines 276-279: all the quoted percentage values in the text do not match with the values plotted in Figure 4b, and appear to be wrong.

21. Figure 5: could the authors include dashed lines in the horizontal and vertical in both plots, to help the reader define the values of data?

22. line 295-301: multiple references in text to Figures S3 and S4, whereas the Supp Info labels them as Figures S5 and S6.

23. lines 307-309: the quoted figure of 18 incidents in the text does not match with (a) the plotted values in Figure 8d which sum to 5 incidents, and (b) he numbers quoted in line 327.

24. line 308: the quoted percentage is incorrect: 18 incidents in 246,690 intersections = 0.007%

25. line 311: the 334,944 ship tracks mentioned in the text does not match the total number of ship tracks in the legends in Figure 2a-d.

26. line 311: closing parentheses is missing after 'Figure 2'.

27. lines 314-319: Figure 8a-b indicate three times more Arctic cyclones defined by wave height, versus near-surface wind speed, whereas the data in Figures 6 and 7 indicate there are slightly more Arctic cyclones using wind speed, than wave height. Could this confusion be explained in text?

28. line 336: "only very small"

29. line 343: "As there is are a"

30. line 356: the reference to tables S2 and S3 here seems inappropriate. The two tables are provided as evidence that the most intense cyclones produced no damage incidents, but the tables contain data for all cyclones, not the five most intense.

31. line 376: the percentage of incidents to intersections is quoted as less than 0.0001%, but it is actually 0.007%, as mentioned earlier.

32. line 378: "but there number" should be "but the number"

33. lines 378-379: the "number of shipping incidents has remained fairly stable between 2010 and 2016" is not consistent with Figure 4a showing the number of incidents growing.

34. lines 385-389: the quoted number of shipping incidents per year, and the trend in the percentage, do not match with the data presented in Figure 4a-b.

35. line 411: the percentage is wrong: 248 incidents with 176,961 ships is 0.14%.

36. line 419: the quoted percentage is wrong.

**Reference**

Coumou, D., Di Capua, G., Vavrus, S. et al. The influence of Arctic amplification on mid-latitude summer circulation. Nat Commun 9, 2959 (2018). https://doi.org/10.1038/s41467-018-05256-8

---

## Author Comment (AC1)

**Response to Reviewers**

**Manuscript titled:** *The Risk of Synoptic-Scale Arctic Cyclones to Shipping*

**Authors:** Alexander F. Vessey, Kevin I. Hodges, Len C. Shaffrey and Jonathan J. Day

**Format of Response**

The comments from each Reviewer are copied (in **GREY TEXT**) and are addressed in turn. Reponses are written in **RED TEXT**. Some comments are similar between the two reviewers, and it is indicated if the response is addressed when responding to the other reviewer. Where text has been edited in our manuscript, this added text is included within this response.

**Response to Reviewer #1**

**Reviewer #1's Overall Comment**

*This is an interesting and potentially important paper because of its creative integration with open access data repositories, addressing transdisciplinary questions with societal relevance, and it should be published as an exemplar. The integration of Satellite Automatic Identification System (S-AIS) data to interpret incidents with maritime ship traffic in relation to sea-ice distributions and cyclonic weather events is innovative with the intersecting dynamics of biogeophysical and socioeconomic systems north of the Arctic Circle. The following comments are intended to enhance the publication with appreciation to the authors.*

**Note to Reviewer #1 - *Paul Arthur Berkman**

The authors thank Reviewer #1 for taking time to read and review this manuscript. We thank Reviewer #1 for expressing their positive review of our manuscript and recommending it for publication.

Please see the point-by-point responses below.

**Response**

1. The Introduction presents well-reasoned conclusions about Arctic maritime ship traffic as a starting point to formulate the questions that stimulated this research.

Thank you for expressing that our introduction is well-reasoned and that our research questions are clearly identified.

2. The Methods includes five subsections that are fine. However, a cross-cutting subsection should be added to clarify the transdisciplinary integration approach. More specifically, this additional subsection should provide details about the mapping methodologies and geospatial analyses with Big Data.

Currently, the methodology section is long and is split into five sub-sections. It was intended that Section 2.5, titled *'Intersecting Past Arctic Ship Tracks with Intense Cyclone Tracks'*, was the section in the Methodology where we describe how we combine the Arctic ship track data and Arctic shipping incident data with the Arctic cyclone track data. Each of these types of data are described in their own sub-section in previous paragraphs. In our view, this concisely describes this transdisciplinary approach of combining data from different sources.

3. In Methods subsection 2.1: The application of sea-ice data from the Hadley Centre in the United Kingdom complements previously published applications of sea-ice data from the National Snow and Ice Data Center in the United States. Noting the data integration

includes the same S-AIS data in Berkman et al. (2020, 2022), how do ship-ice patterns or trends compare with data from the two sea-ice repositories?

Yes, the Arctic sea ice data that is used in our manuscript is different to that described In Berkman et al. (2020, 2022). In our manuscript, we use HadISST2.0, whereas Berkman et al. (2020, 2022) uses data from the National Snow and Ice Data Center (NSIDC).

When conducting our analysis, we found literature that already compared Arctic sea-ice distribution and trends between the HadiSST2.0 data, and other sea-ice data products, such as that from the NSIDC. For example, Comiso et al. (2017) compared HadISST2.0 data with the NSIDC sea-ice data product (among other Arctic sea ice data products), which is used by Berkman et al (2020 and 2022).

The differences between the HadISST2.0 and NSIDC sea ice products are shown in the Figures below, which are from Comiso et al. (2017). Overall, there are very limited differences between Arctic sea ice extent (albeit some differences in sea ice concentration) (Figure 1) and sea ice extent trends (Figure 2) between HadiSST2.0 and NSIDC. As the differences between HadISST2.0 and NSIDC shown in Comiso et al. (2017) are very small, we still think that it is appropriate to still use the HadiSST2.0 data in our manuscript.

[Figure]

Figure 1. Arctic sea ice concentration maps during the JJA between 1979-2015 for (b) Cavalieri et al. (1996) and d) HadISST2. *[SOURCE: Comiso et al. (2017)]*.

[Figure]

Figure 2. Monthly anomalies in (a) ice extent and (b) ice area for SB2, NT1 [*(Cavalieri et al., 1996)]*, HadISST2, and OSI-SAF data sets using passive microwave data from November 1978 to December 2015. *[SOURCE: Comiso et al. (2017)]*.

But to clarify that there are multiple sea ice products available and to quantify the differences between various sea ice data product in our manuscript, we have added the following additional text at line 119.

**[LINE 199]**

*"There are various Arctic sea ice data products available, from various institutions, and Berkman et al. (2020a, 2022) had previously only used Arctic sea ice data from the National Snow and Ice Data Center (NSIDC - Fetterer and Windnagel., 2017). However, Comiso et al. (2017) found that the historical Arctic sea ice extent and trends from HadiSST2.0 are very similar to that of other sea ice datasets, such as the NSIDC (Fetterer and Windnagel., 2017)."*

*References:*

*Comiso, Josefino C., Walter N. Meier, and Robert Gersten. "Variability and trends in the Arctic Sea ice cover: Results from different techniques." Journal of Geophysical Research: Oceans 122.8 (2017): 6883-6900, https://doi.org/10.1002/2017JC012768*

4. In Methods subsection 2.2: "Arctic cyclones are identified in hourly ERA-5 data." As a data integration opportunity, it would seem helpful to mention the relative rates of S-AIS (e.g., satellite orbit) and sea-ice (e.g., daily) data production from the relevant repositories in relation to hourly ERA-5 data.

Yes, we agree with the reviewer. We have added the following text to line 123.

**[LINE 123]**

"Arctic cyclones are identified in hourly ERA-5 data**, which is at a different temporal resolution than the sub-hourly Arctic shipping data and the 6-hourly Arctic sea ice data**."

5. Ships and storms have tracks. It would seem helpful to be consistent with terminology throughout the paper about "ship tracks" and "storm tracks."

Yes, we have made sure to be make this consistent throughout our manuscript.

6. "The Arctic Circle (66.3oN) is incorrect." To be corrected throughout the paper: the Arctic Circle is either 66o33'47.5"N or 66.5oN?

Yes, the reviewer is correct here. We have changed "66.3°N" to "66.5°N" throughout our manuscript.

7. **In Methods subsection 2.3:** "The sensitivity of these thresholds is also assessed in this study, and the number of ships intersected by cyclones with 10 m wind speeds greater than 25 ms−1 (Beaufort Wind Scale 10 and higher) and significant wave heights greater than 4 m (Douglas Sea State Scale 6 and higher) is also assessed." **However, the details are in subsection 2.5.** "In this study, an intersection between a ship track and a cyclone occurs if the ship track is within 3∘ radius of the maximum 10 m wind (greater than 17 ms−1) or significant wave height (greater than 2.5 m)." Additionally, what is a 3o radius in relation to distance?

Thank you for pointing out this repetition. We have deleted the repeated text in subsection 2.5.

3° is equivalent to ~333 km. This has been added to throughout our manuscript. Where we refer to "3°", we have changed the text to "3° (approximately 333 km)".

8. In Methods subsection 2.5: "The number of past ship tracks intersected by an intense cyclone is quantified to determine the number of ship tracks impacted by past cyclones." How? This is the crux of the matter with the missing subsection that is noted above.

Yes, we understand how this could be confusing from our previous manuscript.

The methodology is as follows. The distance between the ships' longitude and latitude coordinates and the longitude and latitude coordinates of the cyclones' maximum 10 m wind speeds and significant wave heights is calculated. An intersection occurs if this distance is less than 3° (i.e., 333 km), and the cyclones' intensity is greater than the 17 ms$^{-1}$ 10 m wind speed and 2.5 m significant wave height thresholds.

To make this clearer in our manuscript, we have edited Figure 10 in our previous manuscript, which is now Figure 8 in our new manuscript. Please see subfigures a) – c) from the old Figure 10 and new Figure 8 below. (Only the first panel from each figure is shown below, for brevity).

**OLD Figure 10.**

[Figure]

[Figure]

[Figure]

*Figure 10. The tracks of the most intense (by significant wave height including tide and swell) Arctic cyclones to occur over the Barents Sea (between 20-30◦E and 71-77◦N) from 2009 to 2010 according to ERA-5. Each row shows the track of each cyclone from the most intense to the 5th most intense. The cyclone 850 hPa relative vorticity centre is denoted by the red cross, and its track by the red line. The location of the maximum significant wave height within a 5◦ radius of the cyclone centre is denoted by the blue marker, with the number of ships intersecting within 3◦ of this location also given. Significant wave height is given by the contoured colours, and the sea level pressure is given by the grey contours. Arctic sea ice extent from the HadISST2.0, where sea ice concentration is >15% is indicated in white. The black crosses denote the AIS satellite positions of ships operating in the Arctic at this time.*

**NEW Figure 8.**

[Figure]

[Figure]

[Figure]

*Figure 8. The tracks of the five most intense Arctic cyclones with the highest significant wave heights over the Barents Sea (between 20-30ºE and 71-77ºN) from 2009 to 2010 based on ERA-5. The cyclone 850 hPa relative vorticity centre is denoted by the red cross, and its track by the red line. The location of the cyclone's maximum significant wave height is denoted by the orange marker, with the 3º (approximately 333 km) radius denoted by the orange dashed line. Significant wave height is given by the colour contours, and the sea level pressure is given by the grey contours. Arctic sea ice extent (sea ice concentration >15%) is indicated in white. The*

*black + crosses denote the AIS-derived ship coordinates at each timestep, with the ships intersected by the cyclone being denoted by the black x crosses.*

Following comments from Reviewer #2 however, this intersection methodology has been revised. Please see the response to their 5[th] comment in their 'Major Revisions' section below.

9. Tables and Figures:

   a. With the five or six Methods subsections, it would seem reasonable to create one synthesis figure for each subsection. Currently, there are 10 figures, and this number should be reduced in the main paper and any ancillary figures could be part of the Supplement.

Given that this manuscript (and the supplementary material) already contains quite a few figures and tables, we have decided not to add synthesis figures for each of the five methodology sections. In our opinion, the Methods sections provide a sufficient explanation of this manuscripts methods as they are.

   b. Conversely, the Tables about shipping incidents in the Supplement are new and should be included in the main paper.

Yes, we agree with the reviewer here. We have moved the tables on past shipping incidents from the supplementary material into the main manuscript. These tables have been added to the *Section 7. Number of Ships Intersected and Impacted by Past Intense Arctic Cyclones*.

   c. Reducing the number of figures may require the Methods subsections to be reorganized in parallel to the story that will be woven through the discussion, introducing Arctic Shipping Data earlier, perhaps as subsection 2.1. The logic of the paper would be tightened by such symmetry.

Yes, this is a good suggestion. The start of our Results section describes trends in Arctic shipping data, so to ensure symmetry in our Methods section, we have moved the subsection

that introduces the Arctic Shipping Data to the start of our Methods section (i.e., as subsection 2.1).

d. It would seem helpful to generate a single synthesis figure with seasonal (left to right and spell-out in the legend) December-January-February (DJF), March-April May (MAM), June-July-August (JJA) and September-October-November (SON):

    I.     "ship tracks";

   II.     "ship density" (noting "ship tracks" is based on individual ships per time and "ship density" is based on all ships per area per time);

  III.     Arctic cyclone wind speed;

  IV.     Arctic cyclone wave height.

We agree with the Reviewer that our manuscript may have too many Sections (previously ten). To make our manuscript more compact and concise, we have combined *Section 7. Frequency of Intense Arctic Cyclones* with *Section 4. Seasonality in Arctic Ship Tracks*. We have also followed the advice of the reviewer to create a synthesis figure of seasonal ship tracks and Arctic cyclone tracks (this new figure is shown below). This has helped us to reduce the number of Figures in our manuscript from 10 to 8.

[Figure]

**Figure 3.** a) - d) All ship tracks from 2010 to 2016 per season (red lines), e) - h) ship track density from 2010 to 2016 per season per grid box (2.0°N x 5.0°E), i) - l) Arctic cyclone track density from 1979 to 2021 per season per unit area (5° spherical cap) of cyclones with 10 m wind speeds greater than 17 ms$^{-1}$, and g) - h) with significant wave heights greater than 2.5 m. a), e), i) and m) - winter (DJF), b), f), j) and n) - spring (MAM), c), e), k) and o) - summer (JJA), and d), h), l) and p) - autumn (SON)). Ship track densities are smoothed using a Gaussian filter equal to 1.0. Mean HadISST2.0 Arctic sea ice extent greater than 15% over each period is shown in white. The solid black line indicates the Arctic Circle (66.5°N).

**e.** In Figure 3: It is interesting that the annual ship-track trends for the Northwest Passage and Northern Sea Route are nearly parallel from 2010-2016, with relative increases in 2014 and 2016. This complements annual ship-ice interactions across latitudes shown in Berkman et al. (2020, 2022).

Yes, in our manuscript, it is shown that annual ship-ice interactions as shown in Berkman et al. (2020, 2022), who suggests that as sea ice extent declines, the number of ship sin the Arctic increases.

This is shown perhaps shown most explicitly however, in Figure 1, where the number of ships in the Arctic is highest in late summer and early autumn months, when Arctic sea ice is typically at its minimum extent. This has been previously shown by Berkman et al. (2020 and 2022).

*This consistency has been added in text in Section 3 "Trends in Arctic Shipping and in Arctic Cyclones"*, with the additional text copied below, with the **BOLD** text indicated the added text.

"The number of Arctic ships per month varies seasonally, with changes in Arctic sea ice extent, which is also shown by Berkman et al. (2020b, 2022). The maximum number of ships in the Arctic generally occurs in the late summer and early autumn months when Arctic sea ice is typically at its annual minimum extent (Figure 1). The minimum number of Arctic ships generally occurs in winter months (Figure 1). For example, in 2012, Arctic sea ice extent was 15.2 million km$^2$ in March but had reduced to 3.6 million km$^2$ in September (National Snow & Ice Data Centre, 2023). **So, the number of ships operating in the Arctic appears correlated with Arctic sea ice extent, where lower sea ice extent coincides with a higher number of ships operating in the Arctic. This is consistent with Berkman et al. (2020, 2022).**"

**10.** Discussion: A paragraph or two should be added about the specific creative applications in this paper as well as generalized utility of open-access data repositories to empower open-ended inquiry, notably in view of risk analyses that are the focus of the journal.

Please see the response to the next comment, as this comment has been addressed there.

**11.** Discussion: Lastly, the audiences for this paper include the decision-making institutions that have remits to address maritime ship traffic in relation to the biogeophysical

dynamics of the Arctic Ocean. It would seem relevant to mention or specify the binding international agreements that address safety of life at sea, search and rescue, pollution prevention, and marine ecosystem impacts.

Yes, we agree with the reviewer here that the applications of this research could be more emphasised within the Discussion section. To address the reviewers 10th and 11th comment, the following paragraph has been added as the final paragraph of our manuscript in *Section 9. Conclusions*. This paragraph aims to emphasise the usefulness of complete open-access data, to call for more open-access complete data repositories. For example, the shipping incident data used is only available from 2005-2017, and the Arctic ship track data used is only available from 2009-2016. This paragraph also aims to emphasise to which industries this study may be useful for.

*"This study exemplifies the capabilities of open access risk analysis and quantifies the risk of past Arctic cyclones impacting Arctic shipping, and also the number of past shipping incidents caused by the passage of an intense Arctic cyclone, which could be useful for decision-making institutions, the insurance industry, and the public. This study relies on open-access atmospheric, shipping tracks and shipping incidents data repositories. Whilst there are considerable amounts of freely available atmospheric data available from various institutions, open-access social data such as ship tracks and shipping incidents is much scarcer. Such social data is even often privatised. Consequently, this study was limited to investigating the risk of Arctic cyclones to shipping in such a short period of history and between 2010 to 2016. The establishment of the Polar Code in 2014 by the International Maritime Organization shows that regulatory authorities expect the number of ships in the Arctic to increase in the following years. As global warming continues to rapidly change the Arctic, extensive and up-to-date ship track and incident data needs to be more publicly available, so that the risks to shipping can be monitored and ultimately mitigated."*

**Response to Anonymous Reviewer #2**

**Reviewer #2's Overall Comment**

*Vessey et al. describe recent trends in ship traffic within the Arctic Circle, and measure the risks they face due to severe weather. The article contains a clear description of the aims of the investigation, its wider context, data and methods used, and explanations of results. It delivers new information on the dangers posed by severe weather to Arctic shipping, of particular benefit to those with exposure in this sector. However, the article has significant shortcomings, and I recommend a major revision before publication. I ask the authors to address the following weaknesses in a revised version of the manuscript: (i) a significant number of grammatical mistakes, (ii) inconsistencies in the values of quantities quoted in the text (often percentages are wrong by a factor 100, and other data in discussions do not match plotted values in figures), and (iii) the proxy of severe weather (a ship within 3 degrees of the location of the maximum wind or wave height above a threshold which itself is within five degrees of an analysed cyclone centre) is unnecessarily complicated, and introduces avoidable error in identifying the true shipping hazard, namely roughness of the sea state at a ship's location.*

**Note to Reviewer #2**

The authors thank Reviewer #2 for reading and reviewing our manuscript and describing it as delivering new information on the dangers that would benefit to those with exposure in this sector. We thank Reviewer #1 for their comments and suggestions, which we believe have improved our manuscript.

Please see the point-by-point responses below.

**Major comments**

1. While the general standard is good, there are many mistakes in the grammar. Some of the more significant errors have been listed in the section 'Minor Comments'. The authors are asked to proofread the entire manuscript for grammatical errors, and meet journal standards.

Thank you for raising these grammatical errors, and we apologise for not identifying them before submission. The lead author of this manuscript is dyslexic, which may have contributed towards some of the grammatical errors.

2. The pdf file has an unusual size of 180 MB and some Apps struggle to load and display a file of this size. A reduction of file size would be useful for the audience. My guess is that Figures 2a-d might be of huge size, because they contain plots of more than 300,000 unique ship tracks. If this guess is correct, then converting them to a different file format may help, or moving Figures 2a-d to Supplementary Information might be suitable?

Yes, Figures 2a-d are very large files. Figures 9 and 10 from our previous manuscript were also fairly large. We have looked at converting these files into a different format to reduce the overall file size of our manuscript. With these corrections, the pdf file size has been reduced from 178 MB to 6.8 MB. The quality of the figures has reduced consequently though, but we believe that the new figures are still legible. We have also removed Figure 9 from our previous manuscript, in response to the reviewer's 5th major comment, which concerns duplicating the analysis for winds and wave heights. We now concentrate on wave heights, hence this revision.

3. All quoted data values in the text need greater inspection by the authors. There is an unusual number of inconsistencies between data values in the text and values plotted in Figures. Some mismatches have been in the 'Minor Comments' section below, but I encourage the authors to comprehensively review this aspect of their manuscript.

Thank you for identifying and pointing out these errors. We have addressed these errors in the 'Minor Comments' section below.

4. There is potential to improve the Introduction.

    a) The opening paragraph of the Introduction describes how sea-ice is declining hence the Arctic is becoming more accessible to shipping. The authors may wish to give a more complete description of Arctic changes in the opening paragraph, before focussing on benefits to shipping. For example, add a sentence or two on its negative aspects both locally (people, wildlife, adapted to colder climate) and on mid-latitude climate – implicated in colder Siberian winter, and droughts/wildfires/floods in mid-latitude summer, e.g. Coumou et al. (2018). Further, the authors may wish to revise their opening statement to reflect how global warming is considered to be the main driver of sea-ice decline. (Multidecadal variability in the climate system may play a minor role.)

We agree with the reviewer here. In our original manuscript, we focus only on the benefits of declining Arctic sea ice extent (i.e., making the Arctic more accessible to shipping). However, there are many consequences of declining Arctic sea ice extent, which can be positive and negative, as the reviewer alludes too. Instead of only concentrating on the benefits to shipping, we agree with the reviewer that we should highlight the various impacts of a warming Arctic, to give the reader an indication of the wider implications of a warming Arctic.

Therefore, we have revised our opening paragraph in our original manuscript. We have added a sentence alluding to other consequences of a warming Arctic, in addition to the benefits to shipping.

The opening two paragraphs have been changed to the following, with additional text shown in **BOLD**:

*"As a consequence of global warming, the Arctic Ocean is becoming increasingly accessible for ships as Arctic sea ice continues to decline (Stroeve et al., 2007, 2012,*

*2014). **Annual mean Arctic sea ice extent has declined from 12.3 million km² in 1979 to 10.5 million km² in 2022, a decline of 15% (Fetterer and Windnagel., 2017). The fastest decline in Arctic sea ice extent occurred in September from 7.1 million km² in 1979 to 4.4 million km² in 2022, a decline of 38% (Fetterer and Windnagel., 2017).** Arctic sea ice is projected to decline further into the future as global surface temperatures are projected to increase further (Stroeve et al., 2012; Wei et al., 2020).*

***This reduction in Arctic sea ice extent could have a detrimental consequences on the Arctic (Serreze and Barry, 2011), and mid-latitude climate systems (Coumou et al., 2018), which may include larger and more frequent Siberian wildfires, stress on local wildlife and ecosystems, and the enhanced release of greenhouse gases into the atmosphere through melting permafrost. However,** reduced Arctic sea ice extent does also provide beneficial opportunities for industries such as shipping, oil exploration, and tourism, which could include shorter journeys between ports in North America, Europe, and Asia (Smith and Stephenson, 2013; Melia et al., 2016, Table 1), access to previously inaccessible natural resources (Harsem et al., 2015), and new destinations for tourism (Maher, 2017). These benefits may lead greater shipping traffic in the Arctic Ocean over the coming decades, consequently increasing the number of ships exposed to extreme weather and other Arctic hazards (Browse et al., 2013; Lasserre, 2014; Melia et al., 2016; Lasserre, 2019)."*

References added to bibliography:

*Serreze, M. C. and Barry, R. G.: Processes and impacts of Arctic amplification: A research synthesis, Global and planetary change, 77, 85–96, 2011.*

*Coumou, D., Di Capua, G., Vavrus, S., Wang, L., and Wang, S.: The influence of Arctic amplification on mid-latitude summer circulation, Nature Communications, 9, 2959, 2018.*

b) The Introduction contains multiple references to information in Table S1. The fact that data in Table S1 is central to this discussion suggests it is more appropriately placed in the main manuscript, either as a Table, or as a map with routes (with polar stereographic map projection).

Yes, Table S1 showing the distances of Arctic sea routes compared to tropical sea routes between some major Ports has been moved to the manuscript. We agree with the reviewer that is Table is central to the discussion and provides evidence of the advantages of using Arctic shipping routes. Therefore, we have moved it from the Supplementary Material and into the main manuscript.

c) My understanding is that the analysis of shipping damage data from the Arctic Council is a key contribution, and an extra paragraph on this dataset in the Introduction would help the reader.

Yes, using shipping incident reports allows us to determine those Arctic ship and cyclone intersections that have resulted in a shipping incident. We have added details of the Arctic shipping incident data to our manuscript, and more specifically in the Introduction Section. The penultimate paragraph has been altered as follows (additional text shown in **BOLD**):

*"The lack of publicly available historic ship track data has been somewhat addressed by Berkman et al. (2020a), who published an open-source Arctic ship track dataset. This contains the transmitted AIS-derived ship location data of ships that travelled north of the Arctic Circle (north of 66.5∘N) but is only available for a limited period from September 2009 to December 2016. Berkman et al. (2020b, 2022) used this dataset and showed that the number of ships in the Arctic has increased between 2010 and 2016. **Arctic shipping incidents from 2005 to 2017 have been collated and made publicly available from Protection of the Arctic Maritime Environment Agency (2023). This database includes incidents occurring due to various causes (e.g., collision, grounding etc.), and describes the ships impacted (e.g., name, tonnage), the incident itself (e.g., if the ship was lost or only partially damaged), and the incidents' consequences (e.g., marine casualty, cargo damage etc.).** Combining past ship tracks and shipping incident reports with past cyclone tracks could provide new insights into quantifying the risk of cyclones to Arctic shipping."*

5. Revision to analysis methods. The authors use the concept of an intersection of Arctic cyclones containing maximum winds/wave heights above a threshold, with the locations

of ship tracks, to identify incidents due to severe weather. This contrasts sharply with the simpler method of identifying severe conditions based on wave height along the ship track. ERA-5 provides all the data required to identify wave height at all locations of a ship track. Could the authors explain why winds are used together with wave heights in a study of ship damage? Do strong winds cause damage, per se? (I thought it was the sea state that damaged modern vessels, but not an expert.) Further, could the authors explain the benefits of including the identification of cyclones here? The definition of severe weather as a ship within 3 degrees of significant winds or waves seems like a proxy for wave height, but given that wave height is available, why use cyclone tracking?

We will address each of the reviewers' comments in turn.

1) Yes, we agree with the reviewer that our intersection methodology is simple, in terms of not determining the significant wave heights at the locations of the ships, even when this information is available from ERA-5. This intersection methodology has been revised considering this comment, and this revision is described below.

The reviewer suggests "identifying severe conditions based on wave height along the ship track", which we have now done, but we still seek to determine if these severe conditions are a consequence of the passage of an Arctic cyclone, instead of from other causes such as shallow/narrow coastlines or from Polar Lows.

So, our new intersection methodology is as follows:
1) The significant wave heights from ERA-5 at the longitude and latitude coordinates of the ship and shipping incident are determined.
2) If the significant wave height at the ship's location is greater than the intensity thresholds (e.g., 2.5 m), it is then determined if this coincides with the passage of a past Arctic cyclone track.
3) If the ship and shipping incident coordinates are within a 3° (approximately 333 km) distance of the the Arctic cyclone's maximum significant wave heights, and an intersection occurs.

This revision to our intersection method is amended throughout our new manuscript section.

2) The reviewer also suggests that we should only show the intersections between ships and cyclones with intense significant wave heights. We agree with the reviewer.

Yes, surface wind speeds and wave heights are related, and high winds tend to cause high waves on the ocean surface. We agree with the reviewer that showing both the intersections between ships and cyclones with intense surface wind speed and wave heights leads to an unnecessary duplication of figures and results in our old manuscript. So, we have also revised this in our new manuscript, and we only show the intersections between ships and cyclones with intense significant wave heights. In our opinion, this makes our manuscript more concise and clearer, and we welcome this revision. We have removed the figures and text describing the intersections relating to wind speed.

In the methodology section, we have also added text to describe why we now concentrate on cyclones with intense significant wave heights, rather than both cyclones with intense wind speeds and wave heights. In Methodology Section titled *"Intersecting Past Arctic Ship Tracks with Past Cyclone Tracks"*, we have added the following text:

*"High surface wind speeds tend to cause tall ocean waves and high significant wave heights. Tall ocean waves are perhaps a more hazardous to ships than high surface wind speeds, as they have a greater ability to make the ship unstable. So, in this study, an intersection between a ship track and a cyclone track occurs if the Arctic ship track longitude and latitude coordinates at the same time step are within $3°$ (approximately 333 km) of the longitude and latitude coordinates of the cyclone's maximum significant wave height, and the significant wave height at the ship's location is greater than 2.5 m. This ensures that the ship is impacted by extreme wave heights and that the extreme wave height conditions are related to the*

*passage of an Arctic cyclone. This intersection methodology is exemplified in Figure 8.*"

3) The reviewer also asks for clarification as to why we use why use cyclone tracking.

Cyclone tracking is important to objective identify past Arctic cyclones, so that the risks of cyclones, which are one of the many hazards to impact shipping in the Arctic, can be discerned. As described in our introduction, there are numerous hazards in the Arctic. These hazards could include cyclones, Polar Lows, drifting icebergs and shallow coastlines. We believe that it is informative to discern which of these hazards have contributed most to reported Arctic shipping incidents. There is very limited literature that attempts to quantify the relative threat of each hazard. We believe that this risk attribution is key to understanding the risk of numerous hazards to human activity in the Arctic, and that this information can be used to help direct hazard assessments and resources to mitigating these hazards and reducing the consequence on human activities. Cyclone tracking is fundamental to objectively identifying past Arctic cyclones and is key to making this manuscript as useful as possible.

It is also important to use cyclone tracking to determine how changes in cyclone climatology and variability impact the risk of Arctic cyclones to shipping. The number of ships operating in the Arctic changing, but the number of intense Arctic cyclones also changes per year. In our study, we show that the year-on-year increase in the number of ship operating in the Arctic is far greater than the int-annual variability in Arctic cyclones.

We have the expertise in cyclone tracking and can investigate and quantify if Arctic cyclones are a dominant cause of Arctic shipping incidents. One of the key and novel results from our manuscript is that Arctic cyclones are found to be only related to a handful of Arctic shipping incidents. And that Arctic cyclones are not the dominant hazard to Arctic shipping in the present-day climate.

We have investigated the Arctic shipping incidents in more detail in our revised manuscript. We have investigated the spatial distribution of the reported Arctic shipping incidents, and it is shown that most shipping incidents have occurred near coastlines rather than over the Arctic Ocean (Figures below).

[Figure]

**Figure 6.** The total number of reported shipping incidents in the Arctic (north of the Arctic Circle) between **a)** 2005 and 2017, and **b)** 2010 and 2016, from the Protection of the Arctic Maritime Environment Agency (2023) database.

This may suggest that over the short time period of between 2005 and 2017, the shallow coastlines found within the Arctic are a greater threat to ships, than Arctic cyclones, which we find only to be related to a handful of shipping incidents. These novel results could be used to spend resources on, for example, surveying the coastlines of the Arctic, rather than spending resources on investigating Arctic cyclones.

So, we believe that the use of cyclone tracking in our manuscript significantly escalates the utility of our manuscript.

We have made edits to our manuscript to help describe and explain why we use cyclone tracking, to make this clearer.

The following paragraph in our Introduction section has been altered as follows (additional text shown in **BOLD**):

*"But, the Arctic is a challenging and hazardous environment for such human activity. Cold temperatures can make working conditions difficult and can cause equipment failures (Larsen et al., 2016), and sea ice can force ships to travel over the shallow and perilous coastlines around the boundaries of the Arctic Ocean (Arctic Monitoring & Assessment Programme: Working Group of the Arctic Council, 2020). Conditions can be made even more dangerous by the passage of a cyclone or a Polar Low, which can cause rough sea conditions due to high winds and high ocean waves (Thomson and Rogers, 2014; Liu et al., 2016; Waseda et al., 2018, 2021). Such conditions could endanger a ship's crew, potentially capsize the ship and its cargo, and cause delays in transit. Arctic cyclones can also enhance the break-up of sea ice (Simmonds and Keay, 2009; Asplin et al., 2012; Parkinson and Comiso, 2013; Peng et al., 2021), which can drive the ice into shipping lanes where it becomes an additional hazard for ships to navigate. **Given the numerous hazards in the Arctic, it is important to assess their relative threat to human activity to inform decision-makers and the public and to ultimately increase the awareness of and resilience against the most threatening Arctic hazards.***"

**Minor comments/corrections**

1. lines 18-19: the sentence beginning 'In September 1980..." provides an indication of the sea-ice decline using September data for 1980 and 2012. I suggest expressing the decline in terms of annual mean Arctic sea-ice over the past 40 years, using data from NSIDC, and expressed in % change (decline) per decade, to create a more robust and representative measure of the decline.

Yes, this is a good suggestion. We had originally chosen to express Arctic sea ice extent decline in terms of the September data, as the decline in Arctic sea ice has been the greatest in this month. But as the reviewer has suggested, using the annual data from NSIDC may be more representative.

So, we have accessed Arctic sea ice data from Fetterer and Windnagel., 2017, and have changed lines 18-19 to include the annual decline in Arctic sea ice extent. Lines 18 to 19 have been changed to:

*"As a consequence of global warming, the Arctic Ocean is becoming increasingly accessible for ships as Arctic sea ice continues to decline (Stroeve et al., 2007, 2012, 2014). **Annual mean Arctic sea ice extent has declined from 12.3 million km$^2$ in 1979 to 10.5 million km$^2$ in 2022, a decline of 15% (Fetterer and Windnagel., 2017). The fastest decline in Arctic sea ice extent occurred in September from 7.1 million km$^2$ in 1979 to 4.4 million km$^2$ in 2022, a decline of 38% (Fetterer and Windnagel., 2017).** Arctic sea ice is projected to decline further into the future as global surface temperatures are projected to increase further (Stroeve et al., 2012; Wei et al., 2020)."*

2. line 21: the word 'however' is not appropriate since there is no contrasting information in this sentence.

'however' has been deleted.

3. the Arctic Circle is defined throughout as 66.3°N, whereas the standard definition is 66°34'N, or approx. 66.6°N (e.g. https://www.pmel.noaa.gov/arctic-zone/faq.html).

Yes, this was also identified by Reviewer #1 and has changed to '66.5°N' throughout the manuscript.

4. line 75: "combined" should be combine?

'combined' has been changed to 'combine'.

5. line 80: grammar error in "has been somewhat been addressed"

The 2$^{nd}$ 'been' has been addressed.

6.  lines 94-96: the description of the sections in the manuscript is not accurate. For example, the summary is given in Section 10, rather than Section 4.

Section 4 has been changed to Section 10.

7.  line 139: error in grammar

This has been changed.

8.  line 169: "Incident south of…" should be "Incidents south of…"

'Incident' has been changed to 'Incidents'.

9.  line 171: "to match the temporal scale" could be replaced by "to match the spatio-temporal scale"

'Temporal' has been changed to 'spatio-temporal'.

10. line 175: "southern quadrant" – either southern half, or southwest quadrant ('southern' is not a quadrant)

This has been changed to 'southern half'.

11. line 197: error in representing the degrees symbol (a superscript '-1')

This has been corrected.

12. lines 201-202: "This is similarly showed by" to "This is similarly shown by"

'Showed' has been changed to 'shown'.

13. lines 204-205: do the authors have any insight into the extent that the increasing number of observed ships is caused by the increasing use of transponders (lines 67-68)? Is it possible to distinguish all large ships over 300 tonnes, or passenger ships, and identify their trend?

Yes, it is possible to distinguish the frequency of larger ships and smaller ships operating in the Arctic from the Berkman et al. (2020) Arctic ship track dataset. Within the dataset, the ship's 'draught' is provided, which indicates the vertical distance between the waterline and the bottom of the ships' hull, and therefore indicates the ship's size and weight. Larger ships tend to have a larger draught as they are heavier and sit lower down in the water.

We have investigated the trend in the number of large and small ships operating in the Arctic between September 2009 and December 2016. This has been achieved by calculating the mean draught of all ships between this period, which is calculated to be 4.55 metres. The frequency per month and year was then calculated of ships with a draught less than (i.e., small ships) and more than (i.e., large ships) 4.55 metres.

Overall, there is a greater increase in the number of small ships per year and month (Figure 1b) between September 2009 and December 2016 than large ships (Figure 1c). However, it is very hard to distinguish, whether this increase in smaller ships is partially due to an increasing use of transponders. But we agree with the reviewer that the increasing use of transponders may artificially lead to an increase in the number of ships operating in the Arctic. However, as the increase in the number of small ships increasing in the Arctic shown in Figure 1 are so great, it is probable that the number of ships has increased in the Arctic from 2010 to 2016.

[Figure]

**Figure 1.** Trends in the frequency of **a)** all ships, **b)** small ships with a draught less than the mean draught across all ships (4.55 m), and , **c)** large ships with a draught more than the mean draught across all ships (4.55 m), with a unique identification number (MMSI) to travel north of the Arctic Circle (66.5°N) per year and month between September 2009 and December 2016 from the Berkman et al. (2020a) Arctic shipping dataset.

To address this uncertainty in whether the number of ships operating in the Arctic has been artificially increased by the increasing use of transponders, we have added Figures 1b and 1c to our manuscript and have added the following paragraph to describe the new results (and sub-figures) and to address this point on uncertainty in the trends.

The following text has been added to Section 3 *"Trends in Arctic Shipping"* to describe this additional figure:

> *"There has been a greater increase in the number of small ships with a draught of less than 4.55 metres from 7,261 in 2010 to 12,193 in 2016 (+68% increase), than the increase in the number of large ships with a draught of more than 4.55 metres from approximately 8,611 in 2010 to 10,117 in 2016 (+17% increase) (Figure 1b and Figure 1c). The draught threshold of 4.55 metres represents the mean draught of all ships that travelled in the Arctic between September 2009 and December 2016. Since 2004, when large ships were mandated to fit AIS transponders, such devices have been increasingly fitted to smaller vessels, and it became mandatory in May 2012 for all fishing vessels with a size greater than 24 metres to have AIS transponders (U.K. Gov., 2014). Such a*

*change in regulation may have artificially increased the number of ships reporting their position when in the Arctic. But, given the increase in the number of ships shown in Figure 1 are so great, and there is a strong increase in the number of large ships that were required to have a AIS transponder from 2004, it is highly likely that the number of ships operating in the Arctic has increased, rather than an increase in the number of ships transmitting their location over this period."*

14. Could the scale in Figures 2e-2h be extended to include low values of track density close to the pole?

We have replotted Figures 2e-2h with an extended scale, to include low values of ship-track density over the pole. But, as ship track density over the pole is so low, the scale must be extended to …, to show these low values of ship-track density close to the pole. So, it is our opinion that we should keep the original scale in the original manuscript.

15. lines 225ff : should references to Figure 2e – f be replaced by Figure 2e – g?

'Figure 2e – f' should be 'Figure 2e – h', and this has been corrected in the manuscript.

16. line 242: reference to a "Table S1.1" needs corrected.

Yes, this has been corrected to 'Table S1'.

17. lines 244-248: the numbers in the text do not match with the plotted values in Figure 3b.

Yes, this has been corrected. Figure 3b shows that approximately 150 ships travelled in the Northern Sea Route (NSR) in 2010.

18. line 261: "only 0.001% of all ships" – inaccurate: 248 incidents among 176,961 ships = 0.14%.

corrected

19. line 273: reference to Figure 1, should be Figure 4a?

No, the reference to Figure 1 is correct here. We are referring to the time series in Arctic shipping number per month, and describing that the highest number of shipping accidents generally occurs in the same month as when there is the highest number of Arctic ships operating in the Arctic.

20. lines 276-279: all the quoted percentage values in the text do not match with the values plotted in Figure 4b, and appear to be wrong.

The percentage value has been changed to 0.1% (35 incidents in 2016 ÷ 34,000 ships operating in the Arctic in 2016 = 0.1%).

21. Figure 5: could the authors include dashed lines in the horizontal and vertical in both plots, to help the reader define the values of data?

Yes, we have added gridlines to this plot.

22. line 295-301: multiple references in text to Figures S3 and S4, whereas the Supp Info labels them as Figures S5 and S6.

This has been corrected.

23. lines 307-309: the quoted figure of 18 incidents in the text does not match with (a) the plotted values in Figure 8d which sum to 5 incidents, and (b) he numbers quoted in line 327.

Yes, this has been corrected.

24. line 308: the quoted percentage is incorrect: 18 incidents in 246,690 intersections = 0.007%

Yes, this has been corrected.

25. line 311: the 334,944 ship tracks mentioned in the text does not match the total number of ship tracks in the legends in Figure 2a-d.

Yes. This has been corrected.

26. line 311: closing parentheses is missing after 'Figure 2'.

This has been corrected.

27. lines 314-319: Figure 8a-b indicate three times more Arctic cyclones defined by wave height, versus near-surface wind speed, whereas the data in Figures 6 and 7 indicate there are slightly more Arctic cyclones using wind speed, than wave height. Could this confusion be explained in text?

This has been corrected.

28. line 336: "only very small"

This has been corrected to 'only a very small'

29. line 343: "As there is are a"

This has been corrected.

30. line 356: the reference to tables S2 and S3 here seems inappropriate. The two tables are provided as evidence that the most intense cyclones produced no damage incidents, but the tables contain data for all cyclones, not the five most intense.

Yes, these supplementary tables describe the ship incidents that have been caused by the passage of all intense Arctic cyclones

31. line 376: the percentage of incidents to intersections is quoted as less than 0.0001%, but it is actually 0.007%, as mentioned earlier.

Yes, this has been corrected.

32. line 378: "but there number" should be "but the number"

This has been corrected.

33. lines 378-379: the "number of shipping incidents has remained fairly stable between 2010 and 2016" is not consistent with Figure 4a showing the number of incidents growing.

Yes this is true. Our consideration was that although the number of shipping incidents has grown from 2010 to 2016, there were very few incidents report in 2016 (35). Figure 4b shows that this increase is mitigated by the faster increasing number of ships operating in the Arctic.

This has been replaced with:
> *"The number of ships in the Arctic has more than doubled from 2010 to 2016, and highest density of ship and intense cyclones in the Arctic occurs over the Barents Sea, around Iceland and over Baffin Bay"*

34. lines 385-389: the quoted number of shipping incidents per year, and the trend in the percentage, do not match with the data presented in Figure 4a-b.

Yes, this has been corrected.

35. line 411: the percentage is wrong: 248 incidents with 176,961 ships is 0.14%.

Yes, this has been corrected.

36. 36. line 419: the quoted percentage is wrong.

Yes, this has been corrected.

---

## Author Response (AR2)

**Response to Reviewers**

**Manuscript titled:** *The Risk of Synoptic-Scale Arctic Cyclones to Shipping*

**Authors:** Alexander F. Vessey, Kevin I. Hodges, Len C. Shaffrey and Jonathan J. Day

**Format of Response**

The comments from the Reviewer are copied (in GREY TEXT) and are addressed in turn. Reponses are written in RED TEXT. Where text has been edited in our manuscript, tracked changes have been included within this response.

**Response to Reviewer #2 – Minor Comments**

**Note to Reviewer #2**

The authors of this manuscript thank Reviewer #2 for taking the time to read and review our revised manuscript, and reviewing it and providing feedback twice. In our opinion, these extra minor comments have helped to elevate this manuscript further, in addition to the comments that Reviewer #2 already suggested in the first round of revisions. We thank Reviewer #2 for expressing their positive review of our manuscript and saying that it "describes high quality research and is a significant contribution to the knowledge in this field". It is very pleasing to hear such a comment. Thank you for recommending our manuscript for publication.

Please see the point-by-point responses to the additional minor comments below.

**Response**

1.  Figure 1: the sum of the values in Figures 1b and 1c should match those in Figure 1a, but they are not consistent. This means that the numbers of ships in 2010 a d 2016 quoted in Lines 220 to 226 are not consistent.

Yes, the reviewer is correct hear. We have updated Figure 1, making sure that the number of ship tracks is consistent between Figure 1a, 1b and 1c.

The updated figure is shown below:

[Figure]

**Figure 1.** Trends in the frequency of **a)** all ships, **b)** all small ships with a draught less than the mean draught across all ships (4.55 m), and , **c)** all large ships with a draught more than the mean draught across all ships (4.55 m), with a unique identification number (MMSI) to travel north of the Arctic Circle (66.5°N) per year and month, from September 2009 to December 2016 from the Berkman et al. (2020a) Arctic shipping dataset.

**2.** Final paragraph of the Introduction: the description of the manuscript is incomplete.

Yes, the reviewer is correct here. Sections 3 – 8 describe the results from this study, and this was not noted in the prior version of the manuscript (only Section 3 was). We have updated this, to give a complete description of the manuscript.

The amended paragraph in the latest manuscript is copied below (we have included the tracked changes, so it is clear where we have removed text):

> The methods used in this study are described in Section 2, including a description of the data and storm tracking method used. In Sections 3 to 8, the results from this study are described, detailing the trends and seasonal spatial distribution of past Arctic ship tracks, past intense Arctic cyclones tracks and past Arctic shipping incidents. The number of ship tracks intersected by passed intense Arctic cyclone tracks, and the proportion of these intersections that resulted in a reported shipping incident is also quantified and described. Finally, a summary of the main conclusions is given in Section 9.

**3.** Section 3: last sentence of the first paragraph: it would be more accurate to write "This shows that the number of ships in the Arctic and transmitting their location has increased between 2010 and 2016."

Yes, we agree with the reviewer here. The amended sentence is copied as below (we have included the tracked changes, so it is clear where we have amended text):

> Between September 2009 and December 2016, 176,961 ships with a unique identification number (MMSI) travelled north of the Arctic Circle (Figure 1a). The number of ships that travelled in the Arctic increased year-on-year from 2010 to 2016 (Figure 1a). This is similarly shown by Berkman et al. (2020b, 2022). In 2010, 15,666 ships with a unique MMSI transmitted an AIS location in the Arctic, whereas in 2016, the number ships operating in the Arctic was +122% higher (more than two times greater) and

approximately 34,780 ships (Figure 1). This shows that the number of ships operating in the Arctic and transmitting their location has increased between 2010 and 2016.

4. Section 3: last sentence of the second paragraph: the authors should consider removing the last part: "…, rather than an increase in the number of ships transmitting their location over this period", because there has been an increase in the number of ships transmitting their location.

Yes, we agree with the reviewer here. We have removed the last par of the sentence as suggested. The amended text is copied as below (we have included the tracked changes, so it is clear where we have amended text):

There has been a greater increase in the number of small ships with a draught of less than 4.55 metres from 7,261 in 2010 to 12,193 in 2016 (+68% increase), than the increase in the number of large ships with a draught of more than 4.55 metres from approximately 8,611 in 2010 to 10,117 in 2016 (+17% increase) (Figure 1b and Figure 1c). The draught threshold of 4.55 metres represents the mean draught of all ships that travelled in the Arctic between September 2009 and December 2016. Since 2004, when large ships were mandated to fit AIS transponders, such devices have been increasingly fitted to smaller vessels, and it became mandatory in May 2012 for all fishing vessels with a size greater than 24 metres to have AIS transponders (U.K. Gov., 2014). Such a change in regulation may have artificially increased the number of ships reporting their position when in the Arctic. But, given the increase in the number of ships shown in Figure 1 are so great, and there is a strong increase in the number of large ships that were required to have a AIS transponder from 2004, it is highly likely that the number of ships operating in the Arctic has increased.<s>But, given the increase in the number of ships shown in Figure 1 are so great, and there is a strong increase in the number of large ships that were required to have a AIS transponder from 2004, it is highly likely that the number of ships operating in the Arctic has increased, rather than an increase in the number of ships transmitting their location over this period.</s>

**5.** Conclusions, line 414: the authors have not proven that the number of ships has more than doubled from 2010 to 2016. Instead, they have presented evidence that the number of ships transmitting a signal has more than doubled.

Yes, we agree with the reviewer here. We have updated the manuscript as suggested, and the amended sentence is as follows (we have included the tracked changes, so it is clear where we have amended text):

 The number of ships operating in the Arctic and transmitting their location using AIS transponders has more than doubled from 2010 to 2016, and the highest density of ship and intense cyclones in the Arctic occurs over the Barents Sea, around Iceland and in Norwegian Seas

**6.** Conclusions: this is lengthy and contains some repetition. It would be of much benefit to the reader if its length was reduced by 30 to 50%.

Yes, we agree with the reviewer that the conclusions section is long and contains some repetition. After reviewing the previous manuscript, we have made changes to make it more concise, and we have reduced the word count by approximately 35%.

The amended Conclusion section is as follows (we have included the tracked changes, so it is clear where we have amended text):

[revised manuscript text omitted]